



# AR6 updates to RF by GHGs and aerosols lowers the probability of accomplishing the Paris Agreement compared to AR5 formulations

Endre Z. Farago[1], Laura A. McBride[2], Austin P. Hope[2], Timothy P. Canty[3], Brian F. Bennett[3], Ross J. Salawitch[1,3,4]

[1]Department of Chemistry and Biochemistry, University of Maryland at College Park, College Park, MD, 20740, USA
[2]Science and Technology Corporation, Columbia, MD, 21046, USA
[3]Department of Atmospheric and Oceanic Science, University of Maryland at College Park, College Park, MD, 20740, USA
[4]Earth System Science Interdisciplinary Center, University of Maryland at College Park, College Park, MD, 20740, USA

*Correspondence to*: Endre Z. Farago (efarago@umd.edu) and Ross Salawitch (rsalawit@umd.edu)

**Abstract.** Many Reduced complexity climate models (RCMs) and Earth System Models (ESMs) use prescribed concentrations or Effective Radiative Forcing (ERF) of Greenhouse Gases (GHGs) and tropospheric aerosols as inputs for projections. Revisions to these datasets, made in Chapter 7 and Annex III of the Sixth IPCC Assessment Report: The Physical Science Basis (AR6, 2021) are vital to ensure the accuracy of climate model forecasts. AR6 provided updates to the formulation of ERF for most GHGs and tropospheric aerosols, relative to values in AR5 (2013). In this work, we provide a comprehensive assessment of how the changes to the ERF datasets impact projections of future warming, using our multiple linear regression energy balance RCM, the Empirical Model of Global Climate (EM-GC). We provide an analysis of the rate of human−induced warming (AAWR) between 1974 and 2014, and Effective Climate Sensitivity (EffCS) from the regression to the observation-based historical climate record with ERF datasets predating the AR6 report (which we term Baseline Framework) and AR6 ERF data (AR6 Framework). Probabilistic projections on future warming that consider the uncertainty in the magnitude of climate feedback and ERF from tropospheric aerosols are provided for four policy-relevant Shared Socioeconomic Pathway (SSP) scenarios. We find AAWR within the AR6 Framework to be 0.18 [0.13 to 0.21 ºC decade$^{-1}$, 5−95% range], a slight increase to the values of 0.16 [0.12 to 0.20 ºC decade$^{-1}$] within the Baseline Framework. The central estimate of EffCS is found to be nearly identical between the two Frameworks, but a narrower range is found for the AR6 Framework at 2.29 [1.54 to 3.11 ºC, 5−95% range] relative to 2.26 [1.45 to 4.37 ºC] within the Baseline Framework. We find Equilibrium Climate Sensitivity (ECS) to be 3.24 [1.92 to 5.15 ºC] for the AR6 best estimate of the pattern effect. Our estimates of AAWR, EffCS and ECS are highly consistent with recent studies and observationally constrained CMIP6 model output. Projections of future warming for the AR6 Framework compared to the Baseline Framework show an increase of 0.2 to 0.4 ºC in the end-of century median warming for the SSP scenarios studied. This increase corresponds to a significantly lowered possibility of accomplishing the goals of the Paris Agreement (PA). In particular, the SSP2−4.5 scenario, that is widely considered to be consistent with current climate policies, only offers an 8% chance of accomplishing the PA upper limit of 2.0 ºC warming by the end of the century within the AR6 Framework.



## 1 Introduction

The Paris Agreement (PA), negotiated in 2015, established the goal of limiting global warming to 1.5 ℃ (PA target) relative to the pre−industrial baseline, with an upper limit of 2.0 ℃ (PA upper limit). Forecasting the rise in global mean surface temperature, for various estimates of the future atmospheric abundances of greenhouse gases and aerosols, is important for assessing the feasibility of accomplishing the goal of the PA.

Numerous modeling efforts focus on assessing the feasibility of various climate policies, and the implications of these
policies on various elements of Earth's climate. Earth System Models (ESMs) are the most comprehensive climate models available and are the primary tool of global warming research. Although ESMs are able to compute the atmospheric concentration of $CO_2$ from emissions, most often these models use prescribed time series of atmospheric $CO_2$ (Jones et al., 2016; Lawrence et al., 2016; Meinshausen et al., 2020). For non-$CO_2$ GHGs, ESMs of Phase 6 of the Coupled Model Intercomparison Project (CMIP6), also use prescribed atmospheric concentration trajectories (Meinshausen et al., 2020). While
there is an effort to make ESMs emissions-based for more greenhouse gases (GHGs) (Meinshausen et al., 2024), many ESMs will need to continue to rely on concentration input for many non-$CO_2$ GHGs.

Reduced Complexity Climate Models (RCMs) are used to compute "best estimate" GHG concentration trajectories from emissions for various scenarios, which can serve as an input to ESMs (Smith et al., 2018a; Meinshausen et al., 2020; Meinshausen et al., 2024). RCMs require substantially lower computational resources than ESMs and allow the exploration of
a wide variety of possible emissions scenarios. Outputs from RCMs serve as the basis for projections of atmospheric abundances of $CO_2$, $CH_4$, $N_2O$ and other GHGs that constitute the Shared Socioeconomic Pathways (SSPs) (Meinshausen et al., 2020).

Chapters 10 (Bindoff et al., 2013) and 11 (Kirtman et al., 2013) of the 2013 IPCC Report (IPCC, 2013b) raised the possibility that "some CMIP5 models have a higher transient response to GHGs and a larger response to other anthropogenic
forcings (dominated by the effects of aerosols) than the real world". The tendency of ESMs to warm more quickly than the observed rise in global mean surface temperature (GMST) over the past three to four decades has also been noted for CMIP6 model output (Tokarska et al., 2020b; Zelinka et al., 2020; McBride et al., 2021). Hausfather et al. (2022) recently termed this tendency the "hot-model problem". Nicholls et al. (2021) examined future projections of GMST from a suite of RCMs and concluded that "the most extreme CMIP6 model projections [of GMST] are outside the range of most RCMs' 5−95th
percentiles [of GMST]".

The Physical Science Basis document published in 2021 (IPCC, 2021a), commonly known as the Sixth Assessment Report (AR6), provided important updates to the projections of future abundances of GHGs and aerosols compared to previously published values in the SSP database (Riahi et al., 2017; Van Vuuren et al., 2017; Fricko et al., 2017; Fujimori et al., 2017;





Calvin et al., 2017; Kriegler et al., 2017; Rogelj et al., 2018). Furthermore, AR6 also updated values of the effective radiative
forcing (ERF) due to the major GHGs and aerosols relative to the Fifth Assessment Report (AR5) (IPCC, 2013b).

In this paper, we review the changes to the projections of GHGs and aerosols, as well as the ERF formulations adapted by
AR6, and compare to prior SSP-based projections of the radiative forcing due to GHGs and aerosols (Sect. 2.3). Then, we use
our multiple linear regression energy balance RCM, the Empirical Model of Global Climate (EM−GC, Sect. 2.4), (Canty et
al., 2013; Mascioli et al., 2012; Hope et al., 2017; McBride et al., 2021), to assess the implications of the AR6 updates on
Effective Climate Sensitivity (EffCS, Sect. 3.1), the rate of warming due to human activity (Attributable Anthropogenic
Warming Rate, AAWR; Sect. 3.1), and projected future warming (Sect. 3.2) for a large ensemble of aerosol trajectories. A
major strength of our EM−GC is the ability to provide probabilistic forecasts of the future rise in GMST for each SSP scenario,
based upon the uncertainty in climate feedback and the radiative forcing due to tropospheric aerosols (Canty et al., 2013;
Mascioli et al., 2012; Hope et al., 2017; McBride et al., 2021). Therefore, we conclude the paper by comparing the likelihood
of achieving the target (1.5 °C warming in 2100) and upper limit (2.0 °C) of the Paris Agreement within the two frameworks,
for various SSP scenarios (Sect. 3.2).

Previous projections of the GMST anomaly found using our EM-GC (McBride et al., 2021) show good agreement with
results from other RCMs that participated in the reduced complexity model intercomparison project (RCMIP) exercise
(Nicholls et al., 2020; Nicholls et al., 2021). In particular, projections of GMST from our model showed good agreement with
results from the Model for the Assessment of Greenhouse Gas Induced Climate Change (MAGICC7) RCM (Meinshausen et
al., 2011a; Meinshausen et al., 2011b; Meinshausen et al., 2020), that was used for the computation of GHG trajectories adapted
by AR6. Consequently, the EM−GC simulations presented here should serve as a good reference point for the impact of the
new AR6 ERF time series on projections of the GMST anomaly found using other RCMs.

## 2 Data and methods

### 2.1 Shared Socioeconomic Pathway scenarios

Shared Socioeconomic Pathway (SSP) scenarios are used to represent various future outcomes in the emissions of GHGs and
tropospheric aerosols, as well as aerosol precursors. SSP scenarios are denoted with the nomenclature SSPx−y, where x
represents the identifier of the baseline SSP pathway (1−5) related to mitigation and adaptation, and y is the target radiative
forcing (in W m$^{-2}$) at the end of the century (commonly referred to as headline or nameplate RF) (O'Neill et al., 2014). As an
example, SSP1−2.6 aims to keep anthropogenic ERF close to 2.6 W m$^{-2}$ in the year 2100 under a sustainable socioeconomic
pathway. Different baseline SSP pathways associated with the same nameplate RF represent the notion that various
socioeconomic trajectories can lead to a similar end-of-century RF (O'Neill et al., 2016). AR6 provides a detailed description
of nine Tier 1 and Tier 2 (O'Neill et al., 2016) SSP scenarios (Chen et al., 2021). Of these nine scenarios, we extensively
analyze the following four: SSP1−1.9, SSP1−2.6, SSP4−3.4, SSP2−4.5.



## 2.2 Modeling framework

We assess the impact on global warming projections of changes to the ERF datasets to GHGs and aerosols introduced by AR6. We define two frameworks, a Baseline Framework that represents the state of knowledge prior to AR6, and an AR6 Framework based on ERF data from Chapter 7 and Annex III of AR6 (Forster et al., 2021; IPCC, 2021b; Smith et al., 2021b; Smith et al., 2021a).

For the Baseline Framework, ERF for GHGs is primarily based on GHG concentrations from the SSP database (Riahi et al., 2017; Van Vuuren et al., 2017; Fricko et al., 2017; Fujimori et al., 2017; Calvin et al., 2017; Kriegler et al., 2017; Rogelj et al., 2018) converted to ERF using the formulae from AR5, as described in Sect. 2.5.3. The ERF due to tropospheric aerosols in the Baseline Framework considers six aerosol types: sulfate, mineral dust, ammonium-nitrate, fossil fuel organic carbon, fossil fuel black carbon and biomass burning organic and black carbon based on data from the Potsdam Institute for Climate Research (Meinshausen et al., 2011c), (Sect. 2.5.3). ERF input data for the AR6 Framework is obtained from Annex III and Chapter 7 of the AR6 report (Forster et al., 2021; IPCC, 2021b; Smith et al., 2021b; Smith et al., 2021a) as described in Sect. 2.5.4.

## 2.3 Effective radiative forcing

Effective Radiative Forcing (ERF) is described as the top-of-the-atmosphere (TOA) energy flux difference (in W m$^{-2}$) due to an imposed perturbation (Myhre et al., 2013a; Boucher and Randall, 2013; Sherwood et al., 2015; Forster et al., 2021). ERF accounts for both tropospheric and stratospheric temperature adjustments (Smith et al., 2018b; Forster et al., 2021), and as such is more representative of the impacts of various forcing agents on the GMST anomaly than stratospheric-temperature-adjusted RF (SARF) or instantaneous RF (IRF) (Forster et al., 2021). ERF describes Earth's energy imbalance due to anthropogenic factors such as GHGs and tropospheric aerosols; hence, ERF is connected quantitatively to the anomaly in GMST that is central to projections of global warming.

### 2.3.1 Greenhouse gases

ERF from GHGs can be computed from their atmospheric concentrations (Byrne and Goldblatt, 2014). Formulations of Myhre et al. (1998) based on results from a line-by-line model (Edwards, 1992; Myhre and Stordal, 1997) were the generally accepted method for converting atmospheric concentrations to radiative forcing for GHGs. These parametrizations were adapted by multiple IPCC reports (TAR, AR4, AR5) (IPCC, 2001, 2007, 2013b). Updates to these formulations, based on the Oslo line-by-line model (Myhre et al., 2006) were provided by Etminan et al. (2016) and Meinshausen et al. (2020), with the latter being adapted into Chapter 7 of the AR6 report (Forster et al., 2021). The formulations by Etminan et al. (2016) and Meinshausen et al. (2020) account for the band overlaps between $CO_2$ and $N_2O$, include shortwave forcing from $CH_4$, and provide an update to the water vapor continuum. Both of these formulations have their own respective advantages and limitations, as discussed in Section 2.7 of Meinshausen et al. (2020) and Section 7.SM.1.1 of AR6 (Smith et al., 2021a).





The AR5 and AR6 formulations are used to compute time series of SARF for each GHG, based on time series of historical and future concentrations of GHGs. In the AR5 RF formulations, SARF computed with the formulations of Myhre et al. (1998) was considered to be equal to ERF, based on an analysis of the RF due to $CO_2$ by Vial et al. (2013). Since no similar analyses were available for other GHGs, ERF was considered equal to SARF for all other well-mixed GHGs in AR5 (Forster et al., 2021). In AR6, tropospheric adjustments for $CO_2$, $CH_4$ and $N_2O$ are assessed to be non-zero (see Sections 7.3.2 and 7.3.5.1 of Forster et al. (2021) for further detail). Consequently, the formulations for ERF given by AR6 allow for tropospheric adjustments, whereas the formulations given by AR5 allow for only stratospheric adjustments.

Annex III of the AR6 report (IPCC, 2021b) adapted not only the ERF formulae described in Meinshausen et al. (2020), but also projections for GHG concentrations for various SSP scenarios published in the same paper. Here and throughout, we consider concentration to be synonymous with volume mixing ratio. The Meinshausen et al. (2020) GHG projections are provided using version 7 of the Model for the Assessment of Greenhouse Gas Induced Climate Change (MAGICC7) RCM (Meinshausen et al., 2011a; Meinshausen et al., 2011b; Meinshausen et al., 2020). Prior to the release of the AR6 report, one of the most up-to-date sources for GHG concentrations for various SSP scenarios was the SSP database (Riahi et al., 2017; Van Vuuren et al., 2017; Fricko et al., 2017; Fujimori et al., 2017; Calvin et al., 2017; Kriegler et al., 2017; Rogelj et al., 2018). These concentration projections were computed with version 6.8 of the MAGICC model, based upon emissions data from various Integrated Assessment Models (IAMs) (Riahi et al., 2017), that were also published in the SSP database. Meinshausen et al. (2020) computed the GHG concentrations using the same GHG emissions database with a newer version of the MAGICC model. MAGICC7 includes several updates and changes to certain gas cycles, as well as the enablement of the permafrost feedback module of the model, which results in additional emissions of $CO_2$ and $CH_4$ from the melting of permafrost.

Figure 1 shows the time evolution of the concentration (Fig. 1a−c) and ERF (Fig. 1d−f) for $CO_2$, $CH_4$ and $N_2O$ within the Baseline and AR6 Frameworks, as well as ERF from tropospheric aerosols ($ERF_{AER}$), the total ERF from all GHGs, and ERF from overall anthropogenic activity (Fig. 1g−i). There is a considerable difference between the Baseline and AR6 datasets, particularly for SSP4−3.4 and SSP2−4.5. Concentrations of $CO_2$ are larger within the AR6 Framework by as much as 30 ppm (Fig. 1a) in the year 2100 compared to the Baseline. Projected concentrations of $CH_4$ within the AR6 Framework are much higher for SSP2−4.5 and SSP4−3.4, compared to the Baseline. For the three major GHGs, an increase for the ERF was found in the AR6 Framework for most SSP scenarios, with a particularly high increase for $ERF_{CO2}$ and $ERF_{CH4}$ of SSP2−4.5 and SSP4−3.4 (Fig. 1d−f). This increase can be attributed to a combined effect of the increase in projected GHG concentrations (Fig. 1a−c) and the updated ERF formulations of Meinshausen et al. (2020), as well as updates to the tropospheric adjustments of RF (Smith et al., 2018b; Hodnebrog et al., 2020; Forster et al., 2021).



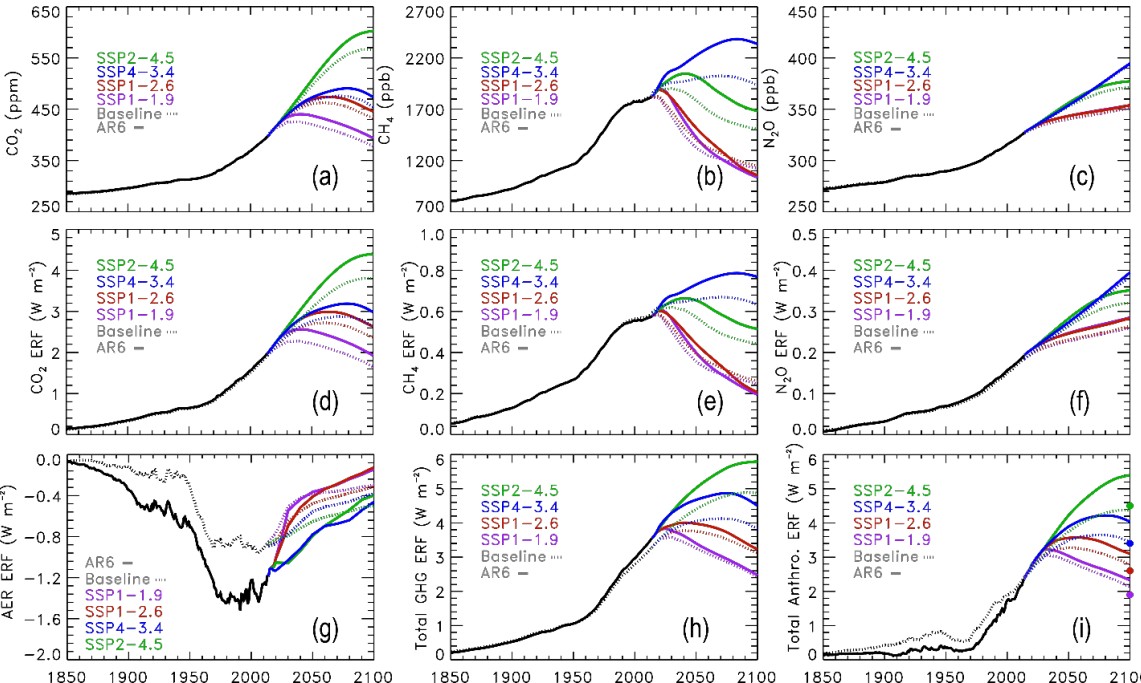

**Figure 1: Concentrations and ERF due to various forcing agents for the Baseline (dashed) and AR6 (solid) frameworks. Colors represent SSP scenarios, as indicated on the individual panels. (a) Concentration of CH₄. (c) Concentration of CO₂. (b) Concentration of CH₄. (c) Concentration of N₂O. (d) ERF from CO₂. (e) ERF from CH₄, including ERF from the enhancement of stratospheric water vapor due to methane. (f) ERF from N₂O. (g) ERF due to tropospheric aerosols. (h) ERF from all GHGs, including halogens and tropospheric O₃, as well as land-use change. (i) Total anthropogenic ERF. Colored circles on the right−hand axis represent the target ERF of each SSP scenario as indicated in the name of the scenario.**

AR6 adapted the SARF formulations of Meinshausen et al. (2020) for $CO_2$, $CH_4$ and $N_2O$, replacing the formulae of Myhre et al. (1998) used in previous IPCC Assessment Reports (Table S1). To convert $SARF_{CO2}$ to $ERF_{CO2}$, an additional +5% tropospheric adjustment was also introduced in AR6 (Forster et al., 2021). This tropospheric adjustment increases $ERF_{CO2}$ by as much as 0.2 $Wm^{-2}$ in the year 2100 for SSP2−4.5. Concentration projections for $CO_2$ in Annex III of AR6 (IPCC, 2021b; Smith et al., 2021b) − which are largely based on Meinshausen et al. (2020) – are also higher than the concentration projections within the SSP Database (Riahi et al., 2017; Fricko et al., 2017; Calvin et al., 2017). Thus, the combination of increased projections on the concentration of $CO_2$ and updates to the conversion of $CO_2$ concentrations to ERF introduced in AR6 result in increased projections of $ERF_{CO2}$ within the AR6 Framework relative to the Baseline, for all four SSP scenarios studied (Fig. 1d).

The inclusion of permafrost emissions of methane result in as much as 50 ppb increase in the atmospheric concentration of methane in 2100, for the scenarios studied in our work (Fig. 3b of Meinshausen et al. (2020)). However, the inclusion of permafrost emissions does not fully account for the difference in the projected concentration of $CH_4$ between the Baseline and AR6 Frameworks (Fig. 1b).

Updates to the atmospheric lifetime of $CH_4$ were introduced in MAGICC7 resulting in a shorter average lifetime for low radiative forcing scenarios, and longer lifetimes for high forcing scenarios (Sect. 2.4.1 of Meinshausen et al. (2020)). This





update, alongside updates to the parametrization of the $CH_4$ cycle in MAGICC7 result in the difference between the projected $CH_4$ concentrations within the Baseline and AR6 frameworks for SSP4−3.4 and SSP2−4.5 depicted in Fig. 1b. In 2100, the atmospheric concentration of $CH_4$ is projected by Meinshausen et. al. (2020) to be higher by 180 ppb and 400 ppb for SSP2−4.5

and SSP4−3.4, respectively, compared to the projections in the SSP database (Riahi et al., 2017; Fricko et al., 2017; Calvin et al., 2017). These findings are consistent with Meinshausen et al. (2020) who concluded "the comparison of mid-century $CO_2$ and $CH_4$ concentrations also reveals that the main reason for higher implied warming of SSP4-3.4 in comparison to SSP1-2.6 are elevated $CH_4$ concentrations".

    While the difference in the projected concentration of $CH_4$ between our two frameworks is considerable for SSP4−3.4 and

SSP2−4.5, the increase in $ERF_{CH4}$ is proportionally smaller in magnitude. The comparably smaller increase in $ERF_{CH4}$ is mostly due to the application of a multiplicative factor of 0.86 used to relate the new SARF formulation of $CH_4$ to ERF. This 14% reduction offsets a significant portion of the would-be increase in ERF due to $CH_4$ caused by higher projected concentrations and spectroscopy−based updates to the formulation of SARF (Forster et al., 2021). Section 7.3.2.2 of AR6 (Forster et al., 2021) assesses the uncertainty in the multiplicative factor used to relate SARF to ERF to be ±0.15, which introduces a sizeable new

uncertainty in the radiative forcing of $CH_4$ for all of the SSPs. Overall, we find a major increase in the $ERF_{CH4}$ within the AR6 Framework compared to the Baseline for SSP4−3.4 and SSP2−4.5, driven by the elevated concentration projections of $CH_4$. In contrast, differences in $ERF_{CH4}$ between the Baseline and AR6 Frameworks for SSP1−1.9 and SSP1−2.6 are quite small. New parametrizations in MAGICC7 were also introduced for $N_2O$ (Meinshausen et al., 2020). These parametrizations result in a minimal increase in the projected concentrations shown on Fig 1c. A small increase in $ERF_{N2O}$ is found between the two

frameworks for all four SSP scenarios (Fig. 1f), mostly attributable to the +7 ± 13% tropospheric adjustment described in Section 7.3.2.3 of AR6 (Forster et al., 2021).

**2.3.2 Tropospheric aerosols and overall anthropogenic ERF**

Tropospheric aerosols (AER) exhibit an overall cooling effect on the atmosphere, both by directly reflecting and absorbing incoming radiation (aerosol direct effect), and through interactions with clouds (aerosol indirect effect) (Forster et al., 2021).

ERF from these two factors is usually denoted $ERF_{ari}$ and $ERF_{aci}$, respectively. We use $ERF_{AER}$ to denote the total ERF due to tropospheric aerosols ($ERF_{ari} + ERF_{aci}$).

    In AR5, the best estimate of the $ERF_{AER}$ was assessed to be −0.9 $Wm^{-2}$, for the time period 1750−2011. In AR6, the best estimate of $ERF_{AER}$ was determined to be −1.1 $Wm^{-2}$, between 1750−2019 (Forster et al., 2021). The corresponding [5th] and [95th] percentile ranges were −0.1 to −1.9 $Wm^{-2}$ in 2011 and −0.4 to −1.7 $Wm^{-2}$ in 2019, for the AR5 and AR6 assessments,

respectively. AR6 also provided the assessment of −1.3 [range −0.6 to −2.0 $Wm^{-2}$] for the 1750−2014 period. We use the 1750−2019 time period below.

    Furthermore, AR5 assessed $ERF_{ari}$ and $ERF_{aci}$ to be nearly equal in magnitude (see Fig. 8.15 of AR5 (Myhre et al., 2013a)). A major difference between these two assessments of $ERF_{AER}$ is that, in AR6, it was stated that the cooling due to the aerosol indirect effect ($ERF_{aci}$) is much larger than the cooling due to the direct effect ($ERF_{ari}$). The change of $ERF_{AER}$ by AR6 to a



value that exhibits more cooling, for a time period that is 8 years more recent than the 2011 end year used by AR5 implies a major shift in the shape of $ERF_{AER}$ as a function of time, because tropospheric aerosol loading is believed to have declined over the past two decades due to successful air quality regulations implemented throughout the world (Smith and Bond, 2014; Fu et al., 2021).

The methodology to develop the time evolution of RF due to aerosols in AR6 is partially based on the modeling study of

Smith et al. (2021c). Similarly to GHGs, there is a direct relationship between $ERF_{AER}$ and the atmospheric abundance of various types of aerosols, which is commonly expressed as a relation between the ERF and the emission of aerosols and aerosol precursors. The parametrization used in AR6 assumes that $ERF_{ari}$ scales linearly with the emissions of various types of aerosol precursors (Smith et al., 2018a; Smith et al., 2021c), while the relation between $ERF_{aci}$ and emissions is logarithmic (Carslaw et al., 2013; Ghan et al., 2013; Stevens, 2015; Smith et al., 2018a; Smith et al., 2021c). Smith et al. (2021c) used aerosol ERF

from various CMIP6 models, which were decomposed to the direct and indirect components using the Approximate Partial Radiative Perturbation (APRP) method (Taylor et al., 2007; Zelinka et al., 2014), from which coefficients for the linear and logarithmic relations (for $ERF_{ari}$ and $ERF_{aci}$, respectively) for each CMIP6 model were determined.

Next, Smith et al. (2021c) created a large ensemble of $ERF_{AER}$ time series of various shapes and magnitudes by a Monte-Carlo sampling around the sets of coefficients obtained from the CMIP6 model-based fits. This aerosol ensemble accounts for

the uncertainty in both the historical time evolution and the magnitude of $ERF_{AER}$. This methodology is adapted by Chapter 7 of AR6 for $ERF_{aci}$, as described in Sections 7.SM.1.3 and 7.SM.1.4 of AR6 (Smith et al., 2021a). The median of the ensemble is adapted as the best estimate for the historical time series of $ERF_{aci}$, scaled to match a value of $-1.0$ $Wm^{-2}$ for 1750 to 2005−2014. For $ERF_{ari}$, AR6 uses the same assumption of a linear connection between aerosol emissions and forcing (Smith et al., 2018a; Smith et al., 2021c), but unlike Smith et al. (2021c), also considers emissions of $NH_3$. Additionally, the AR6

report does not use a Monte-Carlo ensemble to derive a best estimate time series for $ERF_{ari}$ (and corresponding coefficients). Instead, AR6 determines coefficients such that the contributions from the individual aerosol precursor species match the values derived from Myhre et al. (2013b), as described in Section 7.SM.1.3.2 of the AR6 report (Smith et al., 2021a). The AR6 methodology provides a single set of "best estimate" coefficients that can be used to convert emissions data to a time series of $ERF_{ari}$ and $ERF_{aci}$, and consequently, $ERF_{AER}$.

In our model framework, the radiative forcing of aerosols is treated in a probabilistic manner that expands upon the single best estimate $ERF_{AER}$ time series that is associated with each SSP. Therefore, the comparison of single $ERF_{AER}$ time series shown in Fig. 1g highlights only a portion of the difference between the Baseline and AR6 Frameworks.

There are two major takeaways from Fig. 1g. First, for SSP2−4.5 and SSP4−3.4, aerosol cooling is much stronger between 2020 and 2060 in the AR6 Framework compared to the Baseline, whereas for SSP1−1.9 and SSP1−2.6, aerosol cooling is

similar or weaker within the AR6 Framework than in the Baseline. In general, for all four SSP scenarios, the rate of decline in aerosol cooling is more rapid within the AR6 Framework relative to the Baseline. Second, if we use the aerosol ensemble member from our probabilistic treatment that corresponds to the best estimate of $ERF_{AER}$ in the Baseline Framework ($-0.9$ $Wm^{-2}$ in 2011 from AR5) as well as the best estimate in the AR6 Framework ($-1.1$ $Wm^{-2}$ in 2019) to compute the overall




anthropogenic ERF for the respective Frameworks (Fig. 1i), the total anthropogenic ERF between 1850 and 2019 is lower in
the AR6 framework compared to the Baseline, whereas the rise in total anthropogenic ERF between 1950 and 2019 is more
pronounced within the AR6 framework (black dotted and solid lines, Fig. 1i). This difference has important implications for
our estimates of the rate of warming due to human activity, which we discuss in Sect. 2.4.

Finally, we highlight how total anthropogenic ERF at the end of the century considerably exceeds, by as much as 0.7
Wm⁻² for SSP2−4.5, the nameplate RF target for each of the four SSP scenarios (Fig. 1i). At the time the SSP scenarios were
constructed, the value of the nameplate RF at the end of the century was designed to be interpreted as SARF (Tebaldi et al.,
2021). The original purpose of the nameplate RF metric was to provide a numerical representation of a certain climate outcome
(O'Neill et al., 2016). As highlighted by Sect. 7.3.1 of AR6 (Forster et al., 2021), "AR5 recommended ERF as a more useful
measure of the climate effects of a physical driver than [SARF] adopted in earlier assessments". Additionally, as described
above, AR5 considered SARF to be equal to ERF, while this is no longer the case for AR6. Thus, in AR5-consistent settings,
the nameplate RF of a given SSP scenario could be interpreted as either SARF or ERF. Lastly, RF for tropospheric aerosols is
computed as ERF directly from emissions in AR6 (Smith et al., 2021a), which leads to additional differences with the end-of-
century design of the original SSPs. As such, we conclude that the end-of-century RF is larger than the nameplate target for
all four SSPs.

## 2.4 Empirical Model of Global Climate

To quantify the impacts of the updates to ERF datasets outlined above on AAWR, EffCS, and projected warming, we use the
Empirical Model of Global Climate (EM−GC) model (Canty et al., 2013; Mascioli et al., 2012; Hope et al., 2017; McBride et
al., 2021). EM−GC uses a multiple linear regression energy balance approach and computes the GMST anomaly ($\Delta T_{MDL}$)
based on anthropogenic ERF and natural factors as shown in Eq. (1).

$$\Delta T_{MDL,i} = \frac{1+\gamma}{\lambda_p}\left(\Delta ERF_{GHG,i} + s \times \Delta ERF_{AER,i} + \Delta ERF_{LUC,i} - Q_{OCEAN,i}\right) + C_0 + C_1 \times SAOD_{i-6} + C_2 \times TSI_{i-1} + C_3 \times$$
$$ENSO_{i-2} + C_4 \times AMOC_i + C_5 \times PDO_i + C_6 \times IOD_i \tag{1}$$

In Eq. (1), $\Delta ERF_{x,i}$ represent the effective radiative forcing from greenhouse gases (GHG), tropospheric aerosols (AER) and
land use change (LUC) compared to a 1750 baseline. The model uses a monthly time grid, with $i$ being the index of a specific
month. The dimensionless scaling parameter, $s$, is used to account for the uncertainty in the ERF due to aerosols as described
in Sect. 2.4.1.

The natural factors considered when modeling the GMST anomaly are Stratospheric Aerosol Optical Depth (SAOD) from
volcanic eruptions, Total Solar Irradiance (TSI), El-Niño Southern Oscillation (ENSO), Atlantic Meridional Overturning
Circulation (AMOC), Pacific Decadal Oscillation (PDO) and Indian Ocean Dipole (IOD), as outlined in Sect. 2.5.5. Similarly
to previous analyses with EM−GC (McBride et al., 2021), SAOD, TSI and ENSO are lagged by 6,1 and 2 months, respectively.
Additionally, EM−GC quantitatively accounts for the export of heat to Earth's oceans ($Q_{OCEAN,i}$ in Eq. 1), as described below.





The dimensionless parameter $\gamma$ in Eq. (1) is defined as the sensitivity of the global climate to feedbacks due to changes in the anthropogenic ERF due GHGs, AER and LUC. $\gamma$ is related to the climate feedback parameter of EM−GC ($\lambda_\Sigma$) as shown in Eq. (2):

$$1 + \gamma = \frac{1}{1 - (\frac{\lambda_\Sigma}{\lambda_p})} \tag{2}$$

Here, $\lambda_\Sigma$ is the sum of all climate feedbacks from factors such as water vapor, lapse rate, surface albedo and clouds. $\lambda_p$ in Eqs.
(1) and (2) corresponds to the response of a black body to a perturbation with no climate feedback present, with a value of 3.2 W m$^{-2}$ ºC$^{-1}$ (Bony et al., 2006). The above mathematical representation is used in lieu of incorporating the Planck feedback into $\lambda_\Sigma$. For the EM−GC simulations shown below, we assume $\lambda_\Sigma$ to be constant over time, which was found to be a very good approximation within EM−GC, as described by Sect. 3.3.6 of McBride et al. (2021).

Values of the regression coefficients $C_0$−$C_6$ in Eq. (1) are computed by minimizing the cost function shown in Eq. (3).

$$Cost\ function = \sum_{i=1}^{N_{MONTHS}} \frac{1}{\sigma_{OBS,i}^2} (\Delta T_{OBS,i} - \Delta T_{MDL,i})^2 \tag{3}$$

Here, $\Delta T_{OBS,i}$ and $\Delta T_{MDL,i}$ represent the observed and EM−GC modeled GMST anomaly for a given month (i), respectively, while $\sigma_{OBS}$ is the 1$\sigma$ uncertainty associated with each temperature observation.

EM−GC simulations quantitatively account for heat being absorbed by Earth's oceans (ocean heat export, OHE). Time−dependent OHE is computed as shown by Eq. (4), where the constant $\kappa$ is the ocean heat uptake efficiency (in W m$^{-2}$
ºC$^{-1}$) as expressed by Eq. (S1). Further details of our model treatment of OHE is given in the Supplement as well as Sect. 2.1 of McBride et al. (2021).

$$Q_{OCEAN,i} = \kappa(\Delta T_{ATM,HUMAN,i} - \Delta T_{OCEAN,HUMAN,i}) \tag{4}$$

Time series of the natural and anthropogenic factors between 1850 and 2019 are used as inputs to compute the coefficients ($C_0$−$C_6$) in Eq. (1). We refer to this step as the training of the model. We define the historical training period as 1850−2019,
and the future SSP−based projection period as 2020−2100, to be consistent with the definition of historical and SSP−based timeframes outlined in Section 7.SM.1.3 (Smith et al., 2021a) and Annex III (IPCC, 2021b) of AR6. Each fit to the historical GMST record (a set of coefficients) can be extrapolated to the future using Eq. (1), with effects of natural forcings zeroed out, allowing EM−GC to provide a time-dependent GMST forecast for each set of regression coefficients. The model also uses three $\chi^2$ goodness-of-fit constraints, as summarized in Sect. 2.7.

Figure 2 shows the EM−GC best fit to the Hadley Centre Climatic Research Unit version 5 (HadCRUT5) (Morice et al., 2021) historical GMST anomaly record for the Baseline and AR6 Frameworks, assuming the IPCC best estimate RF trajectories for tropospheric aerosols shown on Fig. 1g. The top panels of Fig. 2 show the EM−GC modelled best fit (red) to the HadCRUT5 GMST record (black). Other panels represent the contribution to the modelled GMST from anthropogenic activity (Fig. 2c,d), as well as the natural factors of SAOD, TSI, ENSO, AMOC, PDO and IOD included in Eq. (1) (Fig. 2e−l).





The bottom panels of both columns of Fig. 2 show the modeled (red) and measured (black) Ocean Heat Content (OHC), as well as the corresponding measurement uncertainty (blue). Description of the GMST and OHC datasets used to train the model are provided in Sect. 2.5. EM−GC runs simulate the amount of heat exported to Earth's oceans such that the change in modeled OHC matches the change in observed OHC, over the period where observed OHC data are available. For panel (m) of Fig. 2, the modeled OHC time series is initialized to zero in 1850, and the observed OHC (black line, which is an anomaly) is adjusted

to match the model mean over 1955 to 2019. For panel (n), the observed OHC is identical to that shown in panel (m), and the modeled time series is again adjusted to match the observed mean over 1955 to 2019. This final adjustment leads to a non-zero value for OHC in 1850 for the AR6 Framework run.

      We provide the value of the $\lambda_\Sigma$ and $\kappa$ from Eqs. (1) and (4) on Fig. 2a,b and Fig. 2m,n, respectively. The $\chi^2$ goodness-of-fit parameters (Sect. 2.7) for the modelled GMST anomaly ($\chi^2_{ATM}$) and OHC $\chi^2_{OCEAN}$ are also provided on the same panels.

The rate of increase in GMST between 1975 and 2014 due to human activity (AAWR, Sect. 2.6) is provided as the slope of a linear fit to the anthropogenic contribution to the GMST anomaly in this period on Fig. 2c,d. Similar values of AAWR are found for modest (that is, shifts within a decade) changes in the start and end year used to compute AAWR, due to the linearity of the anthropogenic contribution to GMST (orange lines in Fig. 2c−d).



**Figure 2: EM−GC** assessed contribution of natural and anthropogenic factors to warming for the best fits to the HadCRUT5 GMST record within the Baseline (left) and AR6 (right) Frameworks. Fits are shown for the IPCC best estimate of ERF$_{AER}$ (see text) for each Framework. Both simulations use a training period of 1850 to 2019. **(a,b)** Observed (black) and modelled (red) GMST anomaly (ΔT) relative to a pre-industrial (1850−1900) baseline. Values of λ$_\Sigma$ and χ$^2_{ATM}$ for the best-fit simulation are shown at the top. **(c,d)** Contribution of anthropogenic activity to the GMST anomaly (orange). AAWR, computed as the slope of a linear fit (dashed black line) as described in Section 2.6 is shown at the top. The 2σ uncertainty for the slope of the linear fit is also provided (see text). **(e−h):** Influence of TSI, SAOD and ENSO on the GMST anomaly. **(i,j)** Contribution of AMOC to the GMST anomaly. A linear fit between 1975 and 2014 is provided in a similar manner to the AAWR on panels (c,d). **(k,l):** Influences of PDO and IOD on the GMST anomaly. **(m,n)** Observed (black) and modelled (red) ocean heat content (OHC), with blue bars corresponding to the observational uncertainty. Observed OHC shown in this figure is based on the average of five OHC datasets as described in Sect. 2.5.6. The χ$^2_{OCEAN}$ goodness-of-fit parameter, as well as the ocean heat uptake efficiency (κ) are displayed at the top.

For the AR6 Framework, an AAWR of 0.202 ± 0.006 ºC/decade is obtained from the best fit to the HadCRUT5 GMST record. A considerably smaller value for AAWR of 0.167 ± 0.006 ºC/decade is found for the Baseline framework. As noted above, the rise in total anthropogenic ERF since about 1950 is more pronounced within the AR6 Framework compared to the Baseline (Fig. 1i), due to differences in the best-estimate time series of ERF$_{AER}$ between the two frameworks (Fig. 1g). Since



AAWR is based on the slope of the linear fit to the anthropogenic contribution to warming, a higher value of AAWR is found

for the AR6 framework. Here, both uncertainties for AAWR are based on the goodness of the linear fit for a single model run

and are noted only for completeness. A more realistic estimate of the true uncertainty in AAWR is found using the ensemble

runs of our EM-GC, as described in Section 3.1.

The value of 0.202 ºC/decade found within the AR6 Framework for AAWR is consistent with the assessment of 0.19

ºC/decade between 1980−2020 described in Table 2.4 of AR6 Chapter 2 (Gulev et al., 2021; Forster et al., 2023). Furthermore,

AAWR computed within the AR6 framework shows closer agreement − compared to the Baseline framework − with AAWR

derived from a CMIP6 multi-model ensemble, the median of which McBride et al. (2021) found to be 0.221 ºC/decade, with

the $5^{th}$, $25^{th}$, $75^{th}$ and $95^{th}$ percentiles being 0.151, 0.192, 0.245 and 0.299 ºC/decade, respectively. However, the median value

of AAWR for the CMIP6 ensemble, as well as AAWR obtained from numerous CMIP6 ensemble members, are higher than

our regression-based estimate of AAWR using EM−GC. This finding is consistent with several recent assessments finding that

some CMIP6 GCMs overestimate the extent of warming in response to GHG emissions (Tokarska et al., 2020b; McBride et

al., 2021; Hausfather et al., 2022).

Below, we provide a description of the probabilistic approach utilized by EM−GC to account for the uncertainty in

$ERF_{AER}$. We quantify AAWR and EffCS for the whole range of uncertainty in $ERF_{AER}$ and climate feedback in Section 3.1.

**2.4.1 Uncertainty in $ERF_{AER}$ within EM−GC**

Uncertainty in the magnitude of $ERF_{AER}$ is accounted for by performing the above regression for a large ensemble of aerosol

scenarios. We scale a best estimate time series of aerosol ERF ($\Delta ERF_{AER}$ in Eq. (1)) by a series of constant multiplicative

factors ($s$ in Eq. (1), Fig. S1). The best estimate time series is derived from data from the Potsdam Institute for Climate Research

and Annex III of AR6 for the Baseline and AR6 frameworks, respectively; as outlined in Sections 2.5.3 and 2.5.4. We use

$ERF_{AER}$ in a given reference year (hereinafter $ERF_{AER,t}$; where t denotes the reference year) as the identifier of each of the

aerosol ensemble members. The reference year was chosen as 2011 and 2019 for the Baseline and AR6 frameworks,

respectively, to be consistent with the years for which AR5 and AR6 provided a best estimate and likely range for $ERF_{AER}$.

The regression described in Eqs. 1−3 is performed with each member of the aerosol ensemble being combined with a

series of various values of the climate feedback parameter ($\lambda_{\Sigma}$). In total, 400 values of $\lambda_{\Sigma}$ are used together with 400 values of

the scaling factor $s$ (hence $ERF_{AER,t}$), to create a $\lambda_{\Sigma} - ERF_{AER,t}$ grid with 160,000 elements (hereinafter EM−GC grid). Each

element of the grid corresponds to a fit of the historical GMST record, and thus, a unique set of $C_0−C_6$ regression coefficients.

These sets of coefficients are then used to forecast the future GMST anomaly for each grid member, using ERF projections

for GHGs and aerosols from SSPs. These GMST projections are then filtered and weighed as described in Section 2.7 based

on both the goodness of the fit to the historical GMST record, as well as the magnitude of $ERF_{AER,t}$ compared to the best

estimate and likely range of $ERF_{AER}$ prescribed by AR5 and AR6. Lastly, we use the weighted projections to compute

probabilistic forecasts on the future rise of GMST.





## 2.5 Model inputs

### 2.5.1 Temperature data

In this study, we use the Hadley Centre Climatic Research Unit version 5 (HadCRUT5) (Morice et al., 2021) GMST anomaly records for the training of the EM−GC. Temperature anomalies are with respect to an 1850−1900 pre-industrial baseline. Similarly to previous works using EM−GC (McBride et al., 2021), the uncertainty time series given with the HadCRUT4 temperature record are used for the EM−GC simulations in this paper. The input GMST anomaly is shown by the black line in Fig. 2a-b.

### 2.5.2 Shared Socioeconomic Pathways

In this work, we provide EM−GC output for four different SSP scenarios, two from Tier 1 (SSP1−2.6, SSP2−4.5) and two from Tier 2 (SSP1−1.9, SSP4−3.4) of the ScenarioMIP experiment (O'Neill et al., 2016). Three of these SSP scenarios (SSP1−1.9, SSP1−2.6 and SSP2−4.5) are highlighted as significant scenarios both in Annex III of AR6 (IPCC, 2021b) and Table 1 of Meinshausen *et al.* (2024). SSP4−3.4 is considered as a scenario which was closest to a 50% probability of limiting warming below the PA upper limit of 2.0 ºC in 2100, based on the previous work of McBride *et al.* (2021) using the EM−GC 380 model. Higher forcing scenarios like SSP3−7.0 and SSP5−8.5 are not considered here, as our intention is to focus on the SSP scenarios that are consistent with current or potential future climate policies. Scenarios like SSP3−7.0 and SSP5−8.5 fall into the "The World We Avoided" category of Meinshausen *et al.* (2024), who considers this category to be of a lower priority (see Table 1 of Meinshausen *et al.* (2024)).

### 2.5.3 Baseline Framework

The Baseline framework is constructed based on datasets available shortly before the AR6 report was published. Historical concentrations of GHGs between 1850 and 2014 are from Meinshausen et al. (2017), while GHG concentrations between 2015 and 2100 – with the exception of ODSs – are from the SSP database (Riahi et al., 2017; Van Vuuren et al., 2017; Fricko et al., 2017; Fujimori et al., 2017; Calvin et al., 2017; Kriegler et al., 2017; Rogelj et al., 2018) available at https://tntcat.iiasa.ac.at/SspDb/ (last opened: January 8, 2024). Data in these sources are provided on a yearly and decadal grid, 390 respectively, which are interpolated to obtain a monthly time series of GHG concentrations between 1850 and 2100. Concentrations of $CO_2$, $CH_4$ and $N_2O$ are converted to SARF using the formulae of Myhre et al. (1998). SARF attributable to stratospheric water vapor is computed as 15% of $SARF_{CH4}$ based on Myhre et al., (2007). Concentrations of ODSs beyond 2015 are from table 6-4 of the 2018 Ozone Assessment Report (Carpenter et al., 2018), and are converted to RF using radiative efficiencies for these obtained from the WMO (2018). The SSP database does not provide SARF time series for tropospheric 395 $O_3$, hence our input of $SARF_{O3}$ is developed based on Meinshausen et al. (2011c) with the same methodology used by McBride et al. (2021): $SARF_{O3}$ between 1850 and 2005 are obtained from Meinshausen et al. (2011c) through the Potsdam Institute of Climate Research (PICR) website (https://www.pik-potsdam.de/~mmalte/rcps/, last opened: May 21, 2024), while from 2006





onwards, for each SSP scenario, we use $SARF_{O3}$ from a corresponding RCP pathway in Meinshausen et al. (2011c). $SARF_{O3}$ from RCP2.6 and RCP4.5 are used for SSP1−2.6 and SSP2−4.5, respectively. We use $SARF_{O3}$ from RCP2.6 for SSP1−1.9,

while for SSP4−3.4, a new time series was created, using the linear combination of the RCP2.6 and RCP8.5 time series, weighed by the total GHG SARF in 2100. We note that RCP2.6 is sometimes referred to as RCP3−PD (Meinshausen et al., 2011c), including on the PICR website. Consistent with the pre−AR6 formulations, within the Baseline framework, we consider ERF to be equal to SARF for all GHGs (Myhre et al., 2013a; Forster et al., 2021). Finally, ERF due to land-use change between 1850 and 2011 is obtained from Table AII.1.2 of Annex II of the AR5 (IPCC, 2013a), which is interpolated

onto a monthly time grid. Between 2012 and 2100, we used the 2011 value for $ERF_{LUC}$.

The initial time series for the RF due to tropospheric aerosols is developed in a manner that is similar to that described by McBride et al. (2021). Between 1850 and 2005, we sum RF from six different aerosol types (sulfate, mineral dust, ammonium-nitrate, fossil fuel black carbon and biomass burning organic and black carbon) to obtain a time series of the total aerosol direct effect. RF data for each of these aerosol types are from the PICR (Meinshausen et al., 2011c) for RCP4.5, except for sulfate,

where we use the estimate of Smith *et al.* (2011). The PICR datasets for aerosol direct effects are nearly identical until 2005, so the specific choice of RCP4.5 between 1850 and 2004 carries no significance. Time series for aerosol indirect effects between 1850 and 2005 are developed by scaling the time series of the direct effect using methodology described in Section 3.2.2 of Canty *et al.* (2013), as well as Hope *et al.* (2017). The time series for direct and indirect effects are added to obtain the time series of total RF due to tropospheric aerosols between 1850 and 2005. From 2005 onwards, values of total aerosol RF

published in the SSP database are used. In order to obtain a continuous time series, values of total aerosol RF between 1850 and 2005 are scaled such that the value in 2005 matches the total aerosol RF value in 2005 obtained from the SSP database. The time series of RF due to aerosols described above are treated as ERF for the Baseline Framework. The initial $ERF_{AER}$ time series is then scaled to obtain an ensemble of aerosol forcing scenarios, as described in Sect. 2.4.1.

### 2.5.4 AR6 Framework

EM−GC input data for the AR6 Framework are based on the datasets published in Annex III of AR6 (IPCC, 2021b), and the corresponding data repository (https://doi.org/10.5281/zenodo.5705391, last opening: 9[th] January, 2024) (Smith et al., 2021b). Yearly concentrations for $CO_2$, $CH_4$ and $N_2O$ are obtained from this repository, which are based on the time series by Meinshausen *et al.* (2017) and Meinshausen et al. (2020) for the 1850−2014 and 2019−2100 periods, respectively, with a linear transition between the two datasets between 2015 and 2020, as described by Sections 7.SM1.3 and 7.SM.1.4 of AR6 (Smith

et al., 2021a). Similarly to AR6, the atmospheric concentrations are converted to SARF with the formulae of Meinshausen et al. (2020), using pre-industrial concentrations of 278.3 ppm, 270.1 ppb and 729 ppb for $CO_2$, $N_2O$ and $CH_4$ respectively. These values are ice-core based best assessments of pre-industrial concentrations for these GHGs as described in Chapter 2 of AR6 (Gulev et al., 2021). Tropospheric adjustments of +5%, +7% and −14% are used (Smith et al., 2018b; Hodnebrog et al., 2020) to convert SARF to ERF for $CO_2$, $N_2O$ and $CH_4$, respectively, which are based on Section 7.3.2 and 7.SM.1.3.1 of AR6 (Forster

et al., 2021; Smith et al., 2021a). ERF from stratospheric water vapor is computed as 9.19% of methane ERF; our use of this





scaling value was found to have the highest agreement with the values provided in Annex III of AR6. ERF time series for halogenated compounds and tropospheric ozone are directly adapted from the AR6 Annex III datasets. This dataset provides total ERF from both stratospheric and tropospheric $O_3$, which is dominated by the contribution from tropospheric $O_3$ (see note below Table AIII.3 in Annex III of AR6). All of the above input time series are interpolated onto a monthly time grid, and

then summed to provide the $\Delta ERF_{GHG}$ time series in Eq. (1) for the AR6 Framework.

The ERF time series for land-use change is adapted directly from the AR6 Annex III dataset and interpolated onto a monthly time grid to obtain $\Delta ERF_{LUC}$ to be used in Eq. (1). Finally, the initial ERF time series for tropospheric aerosols is also created using the AR6 Annex III datasets by summing $ERF_{ari}$ (direct effect) and $ERF_{aci}$ (indirect effect) for the entire timeframe for each SSP scenario. The sum is then scaled to create an ensemble for $ERF_{AER}$, as described in Section 2.4.1. The resulting

time series are then interpolated to obtain monthly time series for $ERF_{AER}$.

### 2.5.5 Natural factors

EM−GC simulations consider a variety of natural factors alongside the anthropogenic component of warming, as shown in Eq. (1). The input time series for TSI anomalies used in Eq. (1) is constructed from CMIP6 model data between 1850 and 2014 (Matthes et al., 2017), while values for 2015−2019 are obtained from the Solar Radiation and Climate Experiment (SORCE)

(Dudok De Wit et al., 2017). The SAOD input time series from 1850 to 1978 is based on extinction coefficients at 550 nm from 80 ºS to 80 ºN obtained from the Volcanic Forcing Dataset (Arfeuille et al., 2014) that was prepared for CMIP6 GCM runs. For 1979 to 2018, we use a time series of SAOD at 550 nm from the Global Space-based Stratospheric Aerosol Climatology (GloSSAC v2.0) (Thomason et al., 2018). For the earlier time period (1850 to 1978), the extinction coefficients from the Volcanic Forcing Dataset were integrated from the tropopause to 39.5 km, to obtain a globally averaged SAOD,

weighted by the cosine of latitude from 80 ºS to 80 ºN. For the latter time period (1979 to 2018), we calculate globally averaged SAOD from the GloSSAC dataset using cosine-latitude weighting over the same range of latitudes. For the year 2019, level 3 gridded SAOD product from the Cloud-Aerosol Lidar and Infrared Pathfinder Satellite Observations (CALIPSO) (Vaughan et al., 2004) is used to obtain a global average SAOD, which is then offset by the average difference between the GloSSAC and CALIPSO datasets for the period of overlap (2006−2018) between the two datasets, as described in Section 2.2.5 of McBride

*et al.* (2021). The contribution of TSI and SAOD to the GMST anomaly for the best-fit runs of the Baseline and AR6 frameworks are shown in Fig. 2e-f. Values of TSI beyond 2019, which marks the end of the training period, are set to zero, while for SAOD, we use the value from December 2019 for the 2020 to 2100 period.

The ENSO time series used during the training period (1850−2019) is based on Version 2 of the Multivariate ENSO Index (MEI.v2) (Wolter and Timlin, 1993; Zhang et al., 2019). The MEI.v2 dataset provides data starting in 1979. For 1850 to 1978,

a historical extension based on Wolter and Timlin (2011) and the HadSST3 dataset (Kennedy et al., 2011) is used, as detailed in Section 2.2.6 of McBride *et al.* (2021). Input values for ENSO are also set to zero beyond 2019. The contribution of ENSO to the GMST anomaly for the best-fit runs are shown in Fig. 2g-h.





PDO input data are obtained from the University of Washington PDO index for 1900−2018, with values of zero used outside this time period. IOD input for 1850−2019 was created from the 1º x 1º Sea Surface Temperatures (SSTs) from the
Centennial in situ Observation-Based Estimate (COBE) (Ishii et al., 2005), with values of zero assigned beyond 2019. PDO and IOD were found to have little effect on GMST in EM−GC simulations (McBride et al., 2021), but for completeness, we include these factors. The input AMOC time series is based on SST data from HadSST4 (Kennedy et al., 2019) between the Equator and 60 ºN in the Atlantic Ocean, detrended using the magnitude of anthropogenic radiative forcing, then Fourier-filtered to remove frequencies above $1/9$ yr$^{-1}$ following Sect. 3.2.3 and 4.1.2 of Canty *et al.* (2013) and Sect. 2.2.7 of McBride
*et al.* (2021). Further detail regarding model inputs for ENSO, AMOC, IOD, PDO and SAOD is given in Sections 2.2.5−2.2.7 of McBride et al. (2021). The computed contribution of variations in the strength of AMOC to the GMST anomaly for the best-fit runs is shown in Fig. 2i-j.

### 2.5.6 Ocean heat content

For both Frameworks in this paper, the average of five ocean heat content datasets (Levitus et al., 2012; Balmaseda et al.,
2013; Cheng et al., 2017; Ishii et al., 2017; Carton et al., 2018) is used. The uncertainty for the average OHC record in each month is obtained by computing the 1σ standard deviation between the five datasets for a given month, or by using the uncertainty from Cheng *et al.* (2017) for that month, whichever is greater. The uncertainties from Cheng *et al.* (2017) were chosen for this purpose as these are the largest uncertainties of the five datasets. EM−GC simulations normalize modelled OHC (Fig. 2m-n, red line) to 0 in 1850, which is shown along the average observed OHC time series (black line) and the
uncertainty series described above (blue markers) on Fig. 2m-n for the upper 700m of the ocean. EM−GC simulations assume that the upper 700m of the ocean holds 70% of the total OHC (McBride et al., 2021). Section 2.2.8 and Fig. S9 of McBride *et al.* (2021) provide additional detail on the average OHC dataset and the corresponding uncertainty time series.

### 2.6 Attributable anthropogenic warming rate and effective climate sensitivity

We define Attributable Anthropogenic Warming Rate (AAWR) as the rate of change in GMST due to anthropogenic activity
between 1975 and 2014. The period 1975−2014, chosen in accordance with previous work using our model, spans a 40-year period in which GMST rose in a near-linear fashion due to human activity (McBride et al., 2021). AAWR is determined as the slope of a linear fit to the anthropogenic component ($\Delta T_{MDL,anth}$) of global warming, defined below in Eq. (5), between 1975 and 2014. The time series of $\Delta T_{MDL,anth}$ for the best fts of the Baseline and AR6 Framework simulations are shown by the orange lines in Fig. 2c-d.

$$\Delta T_{MDL,anth,i} = \frac{1+\gamma}{\lambda_p}\left(\Delta RF_{GHG,i} + \Delta RF_{AER,i} + \Delta RF_{LUC,i} - Q_{OCEAN,i}\right) \tag{5}$$

Equilibrium Climate Sensitivity (ECS) is defined as the warming after climate equilibrated to a theoretical doubling of the pre-industrial concentration $CO_2$ (IPCC, 2001, 2021c; Forster et al., 2021). Since equilibrium can take centuries to reach due to



the slow heat transfer to deep oceans (Hansen et al., 2011; Church et al., 2013; Tokarska et al., 2020a), often the more short-term Effective Climate Sensitivity (EffCS) (Gregory et al., 2020; Tokarska et al., 2020a; Spencer and Christy, 2023) is used.
EM−GC output is used to provide an estimate of EffCS which is based upon the regression to the historical GMST record. We infer EffCS from the ERF due to the doubling of the pre-industrial $CO_2$ concentration ($\Delta ERF_{2xCO2}$) as shown in Eq. (6), which is consistent with the methodology described in Box 7.1 of AR6 (Forster et al., 2021).

$$EffCS = \frac{1}{\lambda_p - \lambda_\Sigma} \times \Delta ERF_{2 \times CO2} \qquad\qquad (6)$$

This method of computing EffCS depends on the formulation of ERF. Within the Baseline Framework, we use the RF formula
of Myhre et al. (1998), which leads to $\Delta ERF_{2xCO2} = 5.35 \times \ln(2) = 3.71\ Wm^{-2}$. For the AR6 Framework, we use a $\Delta ERF_{2xCO2}$ value of 3.93 Wm$^{-2}$ based on Section 7.3.2.1 and 7.SM.1.2 of AR6 (Forster et al., 2021; Smith et al., 2021a). Thus, EffCS computed using the AR6 ERF$_{CO2}$ formula is 6% larger relative to EffCS calculated with the Myhre et al. (1998) formula for a given value of $\lambda_\Sigma$. Finally, we note that climate sensitivity deduced from historical warming may be different from true ECS, as the historical climate feedback could differ from the climate feedback under an abrupt 4×CO$_2$ forcing scenario that is
often used to evaluate ECS in ESMs (Andrews et al., 2018; Andrews et al., 2019; Winton et al., 2020; Forster et al., 2021).

**2.7 Observational constraining and ensemble weighting**

The 160,000 member $\lambda_\Sigma$−ERF$_{AER}$ grid described in Sect. 2.4.1 is constrained by observational data of the GMST anomaly and OHC, using three $\chi^2$ based metrics (McBride et al., 2021) given by Eqns. S2−S4. Two of these $\chi^2$ metrics quantify how well the modelled GMST anomaly represents the observed temperature anomaly of the atmosphere for the entire training period
(1850 – 2019, $\chi^2_{ATM}$) and the GMST anomaly of the last 80 years (1940 – 2019, $\chi^2_{RECENT}$). The third $\chi^2$ metric ($\chi^2_{OCEAN}$) is a goodness-of-fit value between the observed and modeled ocean heat content. Numerical values of $\chi^2_{ATM}$ and $\chi^2_{OCEAN}$ for the best fit run of the Baseline and AR6 Frameworks are given in Fig. 2a-b and 2m-n, respectively. The $\chi^2_{RECENT}$ metric is used because without this constraint, some solutions with values of $\chi^2_{ATM}$ less than or equal to 2 have a visually poor simulation of the rise in GMST over the past 4 to 5 decades (McBride et al., 2021). Only members of the $\lambda_\Sigma - ERF_{AER,t}$ grid that yield a good
fit to the historical GMST and OHC data are accepted; that is, grid members for which all three $\chi^2$ metrics yield values less than or equal to 2.

   For the results shown in Section 3, the accepted grid members that are considered to be good fits to the observational data are then weighted by the assessed likelihood that their specific value of ERF$_{AER}$ in the reference year 2011 (for the Baseline Framework) or 2019 (for the AR6 Framework) is "true". To derive these weightings, we use an asymmetrical Gaussian
function that is centered around the IPCC best estimate of the ERF of aerosols in the reference year (−0.9 Wm$^{-2}$ in 2011 within the Baseline Framework, and −1.1 Wm$^{-2}$ in 2019 for the AR6 Framework). These weighting functions are shown in Fig. 3a-b. The 2σ boundaries of the Gaussians are based on the boundaries of the likely range of ERF$_{AER}$ in the same year provided by the corresponding IPCC reports (see Table S3). The Gaussians are asymmetrical, as the likely range of ERF$_{AER}$ specified in both the AR5 and AR6 reports is not symmetrical around the respective best estimate. The weighted grid is then used to provide





probabilistic estimates of AAWR and EffCS (Sect. 3.1), as well as probabilistic forecasts on the GMST anomaly (Sect. 3.2), which serve as the primary output of EM−GC.

## 3 Results

### 3.1 Ensemble−based assessment of AAWR and climate sensitivity

In Sect. 2.4, we briefly analyzed the AAWR obtained from EM−GC best fits to the GMST record, assuming the AR5 and AR6
prescribed radiative forcing for aerosols, for the Baseline and AR6 Frameworks, respectively. We found a significant increase in the value of AAWR from 0.167 ºC/decade within the Baseline Framework, to 0.202 ºC/decade for the AR6 Framework. In this section, we evaluate AAWR and EffCS for the entire 160,000 member EM−GC grid, which accounts for the uncertainty in the magnitude of climate feedback and ERF of aerosols (Sect. 2.4.1).

Figure 3 shows AAWR (Fig. 3c−d) and EffCS (Fig. 3e−f) for the EM−GC $\lambda_\Sigma$ − $ERF_{AER,t}$ grid, within the Baseline (left)
and AR6 Frameworks (right). Colors denote the values of AAWR and EffCS found for simulations that satisfy all three $\chi^2$ metrics, as described in Sect. 2.7. Fig 3a−b shows the Gaussians used to convert the $\lambda_\Sigma$ − $ERF_{AER,t}$ grid output into a probabilistic quantification of AAWR and EffCS, which are presented in the form of Probability Distribution Functions (PDFs) in Fig. 4. The height of bars on Fig. 4 corresponds to the probability of AAWR (Fig. 4a) and EffCS (Fig. 4b) − obtained from the Gaussian-weighting of the EM−GC grid − being in the range defined by the width of each column. We use a bin size of
0.005 ºC decade$^{-1}$ and 0.1 ºC in Fig. 4 for AAWR and EffCS, respectively. Figure 4 also shows the EM−GC median for AAWR and EffCS, which we define as the values of AAWR and EffCS corresponding to 50% probability, alongside the 5th to 95th percentile range for AAWR and EffCS.



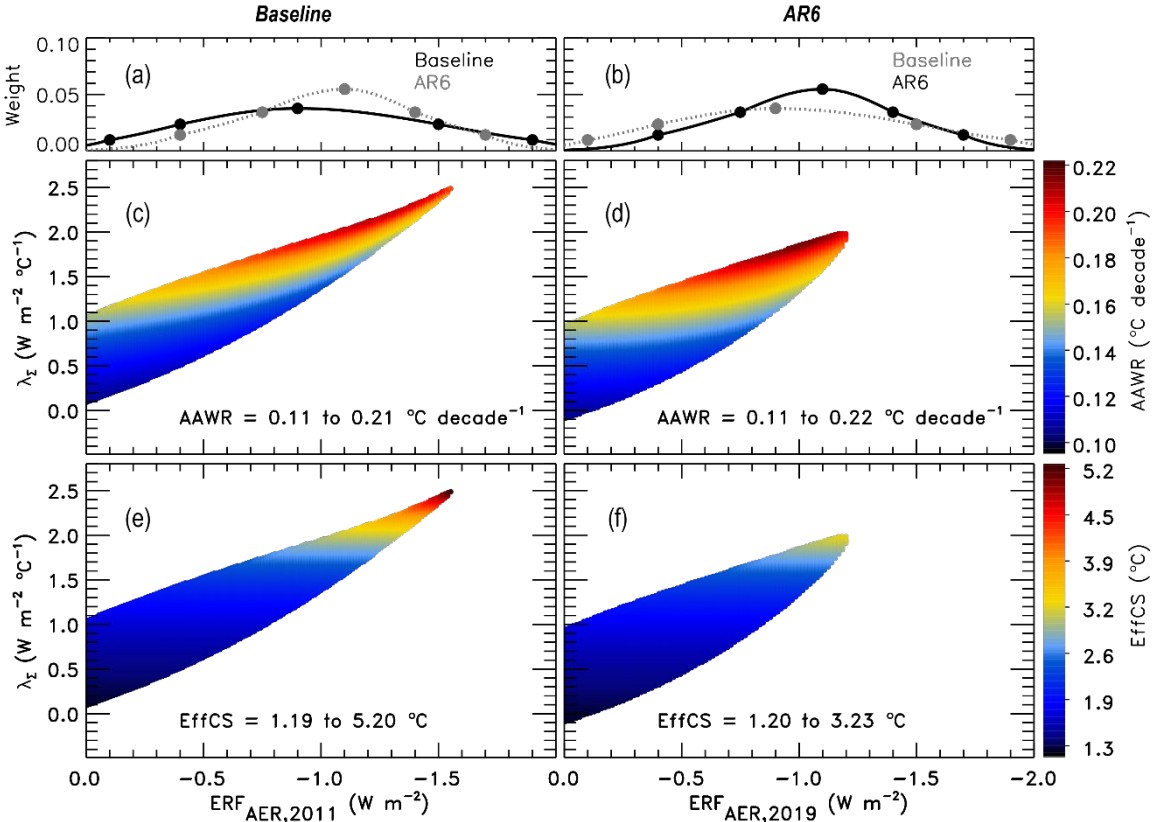

**Figure 3: Aerosol weighting method and EM−GC computed values of AAWR and EffCS for combinations of ERF$_{AER}$ and λ$_{\Sigma}$. (Left):**
**Simulations using the Baseline Framework. (Right): Simulations using the AR6 Framework. (a−b) Asymmetrical Gaussians used to**
**weight aerosol scenarios for probabilistic forecasts, as described in Sect. 2.7. Points marked on the Gaussians represent specific ERF**
**values used as the central values, as well as 1σ and 2σ boundaries of each Gaussian (Table S3). The Gaussians are overlaid for visual**
**comparison. The Gaussians shown with the solid black line are used to weight the EM−GC output in each column. (c−d) EM−GC**
**computed values of AAWR for the λ$_{\Sigma}$ − ERF$_{AER,t}$ grid. Colors denote the specific values of AAWR as indicated by the color bar on**
**the right, and are only shown for the combinations of ERF$_{AER}$ and λ$_{\Sigma}$ for which a good fit to the HadCRUT5 historical climate record**
**was found. (e-f) EM−GC computed values of EffCS for the λ$_{\Sigma}$ − ERF$_{AER,t}$ grid.**

Within the Baseline Framework (Fig. 3c), the historical GMST record can be fit with combinations of high λ$_{\Sigma}$ and strong
aerosol cooling. For example, good fits to the historical GMST record were found with aerosol cooling as negative as −1.6
Wm$^{-2}$ in 2011, which corresponds to much stronger aerosol cooling than the AR5 best estimate of −0.9 Wm$^{-2}$ within the same
year. Conversely, within the AR6 Framework, the strongest aerosol cooling for which a good fit was found (about −1.2 Wm$^{-2}$
in 2019) is very close to the AR6 best estimate of −1.1 Wm$^{-2}$ in 2019. Thus, within the AR6 Framework, it is not possible to
fit the historical GMST record with aerosol cooling much stronger than the corresponding IPCC best-estimate. Within the AR6
Framework, the ERF$_{AER,2019}$ range for which a good fit to the historical GMST record was found is 0.0 to −1.2 Wm$^{-2}$, which
is broadly consistent with the AR6 likely range for ERF$_{AER,2019}$ of −0.4 to −1.7 Wm$^{-2}$ (Forster et al., 2021). While good fits to
the historical GMST record were found with weak aerosol cooling of 0 to −0.4 Wm$^{-2}$, these simulations are given very low
weights (Fig. 3b) because the value of ERF$_{AER}$ lies outside of the AR6−assessed likely range. These weak aerosol cooling





simulations are also associated with values of climate feedback ($\lambda_\Sigma$) that fall outside the "likely" range of total feedback derived from Table 7.10 of AR6 (see Supplement for details). Finally, even though EM−GC simulations can obtain good fits to the HadCRUT5 record with $ERF_{AER}$ being greater than 0 Wm$^{-2}$ and with $\lambda_\Sigma$ being negative, these simulations are discarded (and

not shown in Fig. 3) because Section 7.3.3.4 of AR6 states that it is "virtually certain" that total aerosol ERF is negative. The $ERF_{AER}$ cutoff in Fig. 3c−f is thus set at 0 Wm$^{-2}$.

      The range of AAWR found using the entire EM−GC grid is near identical between the Baseline and AR6 Frameworks at 0.11 to 0.21 ºC decade$^{-1}$ and 0.11 to 0.22 ºC decade$^{-1}$, respectively (Fig 3c−d). However, the median AAWR increases from 0.16 to 0.18 ºC decade$^{-1}$ from the Baseline to the AR6 Framework (Fig. 4a). Similarly to the range of AAWR derived from

the entire $\lambda_\Sigma$ − $ERF_{AER,t}$ grid, the 5$^{th}$ to 95$^{th}$ percentile ranges are found to be consistent between the Baseline and AR6 Frameworks, at 0.12 to 0.20 ºC decade$^{-1}$ and 0.13 to 0.21 ºC decade$^{-1}$, respectively. As described above, AAWR found for the best fit to the GMST record using the IPCC prescribed best estimate for the RF of aerosols is 0.167 ºC decade$^{-1}$ and 0.202 ºC decade$^{-1}$ for the Baseline and AR6 Frameworks, respectively (Fig. 2c−d). For the Baseline Framework, the best−fit AAWR of 0.167 ºC decade$^{-1}$ found using the IPCC best estimate aerosol cooling is close to the EM−GC median of 0.16 ºC decade$^{-1}$.

Conversely, for the AR6 Framework, the median AAWR of 0.18 ºC decade$^{-1}$ is about 10% smaller than the AAWR of 0.202 ºC decade$^{-1}$ shown in Fig. 2d. This difference is caused by the fact that the IPCC AR6 best estimate of −1.1 W m$^{-2}$ for $ERF_{AER,2019}$ is very close to the upper end of the range of $ERF_{AER}$ for which a good fit was found (Fig. 3d), whereas for the Baseline Framework, the AR5 best estimate for aerosol cooling of −0.9 W m$^{-2}$ is approximately in the middle of the range of $ERF_{AER}$ for which good fits to the historical GMST record are found.

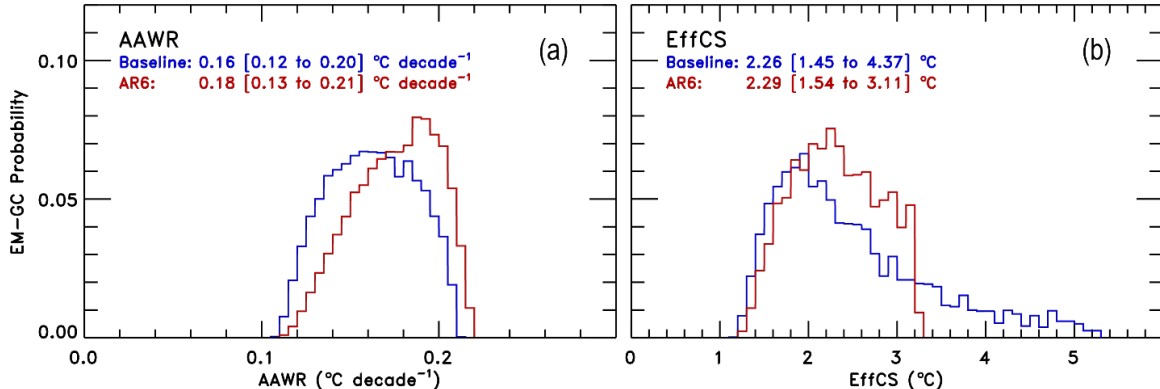


**Figure 4: (a) Probability Distribution Functions (PDFs) for AAWR obtained from EM−GC simulations trained with the HadCRUT5 temperature dataset. AAWR from the Baseline Framework is shown in blue, while AAWR from the AR6 Framework is shown in red. The EM−GC median (defined as the 50$^{th}$ percentile probability), and 5$^{th}$ to 95$^{th}$ percentile range is displayed at the top. (b) PDFs of EffCS within the Baseline (blue) and AR6 Framework (red), with the corresponding median and range.**

The EM−GC computed median and 5−95% range of AAWR are 0.18 [0.13 to 0.21 ºC decade$^{-1}$] between 1974 and 2014 for the AR6 Framework. This range and median are consistent with several other studies, such as Samset et al. (2023), who found a rate of increase in warming to be 0.19 ± 0.01 ºC decade$^{-1}$ between 1973 and 2022 for the HadCRUT5 GMST record. Samset et al. (2023) also found rates for other temperature records such as GISTEMP v4, NOAA v5.1 and Berkeley Earth to



be 0.19, 0.18 and 0.17 ºC decade⁻¹, respectively. Our HadCRUT5−based median of 0.18 ºC decade⁻¹ is consistent with the
analysis of Samset et al. (2023) who employed a variety of historical temperature records, as well as the assessment of 0.19 ºC
decade⁻¹ between 1980−2020 described in Table 2.4 of AR6 Chapter 2 (Gulev et al., 2021; Forster et al., 2023). As highlighted
by Samset et al. (2023) however, warming has accelerated since 1990 at a rate of 0.008 to 0.025 ºC decade⁻¹ per decade
depending on the temperature record studied, with Forster et al. (2023) finding a rate of 0.2 ºC decade⁻¹ for human−induced
warming for the 2013−2022 period, while Ribes et al. (2021) found this rate to be 0.23 ºC decade⁻¹ for the 2010−19 period.

Our median and range of AAWR of 0.18 [0.13 to 0.21 ºC decade⁻¹] within the AR6 Framework is considerably lower
than the median and 5−95% range of 0.221 [0.151 to 0.299 ºC decade⁻¹] derived from a CMIP6 multi-model ensemble by
McBride *et al.* (2021). The CMIP6 median obtained by McBride *et al.* (2021) is greater than the upper limit of the range found
in our study. Samset *et al.* (2023) also found that "virtually all CMIP6 simulations have higher 50−year warming rates than
the observations", using an ensemble of 119 ESM simulations from CMIP6. In particular, Samset et al. (2023) found that
models with ECS being greater than 3.0 ºC tend to quantify the rate of warming to be above 0.2 ºC decade⁻¹ up to about 0.43
ºC decade⁻¹ for certain models with ECS around 5 ºC (Fig. 2d of Samset et al. (2023)). Armour *et al.* (2024) showed that
CMIP5/6 models that tended to over-estimate the observed warming trend over the 1981 to 2014 time period tended to have
larger values of EffCS, which they attribute to gross differences in the observed versus modeled warming patterns, particularly
in the tropical Pacific Ocean. Overall, AAWR found using our AR6 Framework is consistent with other lines of empirical
evidence in the recent literature, and tends to be quite a bit lower than the CMIP6 multi-model mean.

EffCS found using the three $\chi^2$ based metrics has a range of 1.20 to 3.23 ºC in the AR6 Framework relative to a range of
1.19 to 5.20 ºC for the Baseline Framework (Fig. 3e−f). The lower limit is found to be near identical between the two
Frameworks; however, the upper limit of EffCS is substantially smaller within the AR6 Framework. This difference originates
from the fact that the HadCRUT5 GMST record can be fit with combinations of high climate feedback and strong aerosol
cooling only for the Baseline Framework, as described above. Fits with very strong aerosol cooling correspond to high $\lambda_\Sigma$ and
yield high values of EffCS (Eq. (6)) in the Baseline framework, resulting in a high upper bound for the range of EffCS.

The median EffCS is also found to be near identical between the Baseline and AR6 Frameworks at 2.26 and 2.29 ºC,
respectively, with corresponding 5−95% ranges being [1.45 to 4.37 ºC] and [1.54 to 3.11 ºC] (Fig. 4b). The 95th percentile
value of EffCS is considerably smaller within the AR6 Framework relative to the Baseline, due to the aforementioned
phenomenon of not being able to obtain good fits to the GMST anomaly in the AR6 Framework for aerosol scenarios with
ERF$_{AER,2019}$ being more negative than about −1.2 W m⁻². Given that in the AR6 Framework the Gaussian used to weight the
$\lambda_\Sigma$ − ERF$_{AER,t}$ grid is centered around the IPCC best estimate of −1.1 W m⁻², the few combinations of $\lambda_\Sigma$ − ERF$_{AER,2019}$ that
yield good fits for which ERF$_{AER,2019}$ is between −1.1 and −1.2 W m⁻² are still assigned high weights. Within the Baseline
Framework, good fits with strong aerosol cooling scenarios (as negative as −1.6 W m⁻² in 2011) are far from the AR5 best
estimate of −0.9 W m⁻² cooling and are therefore assigned small weights (Fig. 3a). The high cooling scenarios correspond to
the low probability tail of the PDF with EffCS greater than about 3.5 ºC shown in Fig. 4b for the Baseline Framework. The



range of EffCS found for the AR6 Framework is considerably narrower than for the Baseline, which is well represented by the difference in the 95[th] percentiles for EffCS for the two frameworks of 3.11 °C and 4.37 °C, respectively (Fig. 4b).

A recent study by Skeie *et al*. (2024b) found the best estimate and 90% uncertainty range for EffCS to be 2.2 [1.6 to 3.0 °C] using a Bayesian estimation model and ERF datasets from AR6. Our estimate of 2.29 [1.54 to 3.11 °C] within the AR6 Framework exhibits close agreement with this study. Furthermore, Skeie *et al.* (2024b) assume climate feedback to be constant over the historical period, similar to our work. However, climate feedbacks may not be constant over time, partly due to their dependence on the spatial pattern of warming, often termed "pattern effect" (Andrews et al., 2015; Armour, 2017; Stevens et al., 2016; Skeie et al., 2024b). Since climate sensitivity is derived from the magnitude of climate feedback (i.e. (Forster et al., 2021)), the magnitude of the pattern effect governs how much climate sensitivity inferred from historical records (EffCS) differs from ECS. The sensitivity of GMST projections found using our model to time-dependent feedback is quantified in Section 3.3.6 of McBride *et al.* (2021).

Next, we quantify the effect of time-dependent feedback on ECS. Similar to Skeie *et al.* (2024b), we apply the AR6 formulation for the impact of the pattern effect on the ensemble output of the AR6 Framework. The impact of time-dependent feedback on ECS is quantified as in Eq. (7), where α' represents the difference between climate feedback inferred from historical warming and the climate feedback corresponding to that found for an abrupt doubling of the concentration of $CO_2$.

$$ECS = \frac{1}{\lambda_p - \lambda_\Sigma - \alpha'} \times \Delta\mathrm{ERF}_{2 \times CO2} \tag{7}$$

The formulation shown in Eq. (7) is equivalent with the methodology for the computation of ECS with the inclusion of α' described in Section 7.5.2 of AR6 (Forster et al., 2021). The coefficient α' is assessed by Section 7.4.4.3 of AR6 to have a value of 0.5±0.5 W m$^{-2}$ K$^{-1}$, at a low confidence level. Figure 5 shows ECS for values of α' within the AR6 assessed range of 0 to 1 W m$^{-2}$ K$^{-1}$, in increments of 0.1 W m$^{-2}$ K$^{-1}$ (black circles, median), where the error bars represent the 5[th] and 95[th] probabilistic range of ECS. The dashed horizontal line and shaded area correspond to the central value and very likely range of ECS from Table 7.13 of AR6 (Forster et al., 2021).



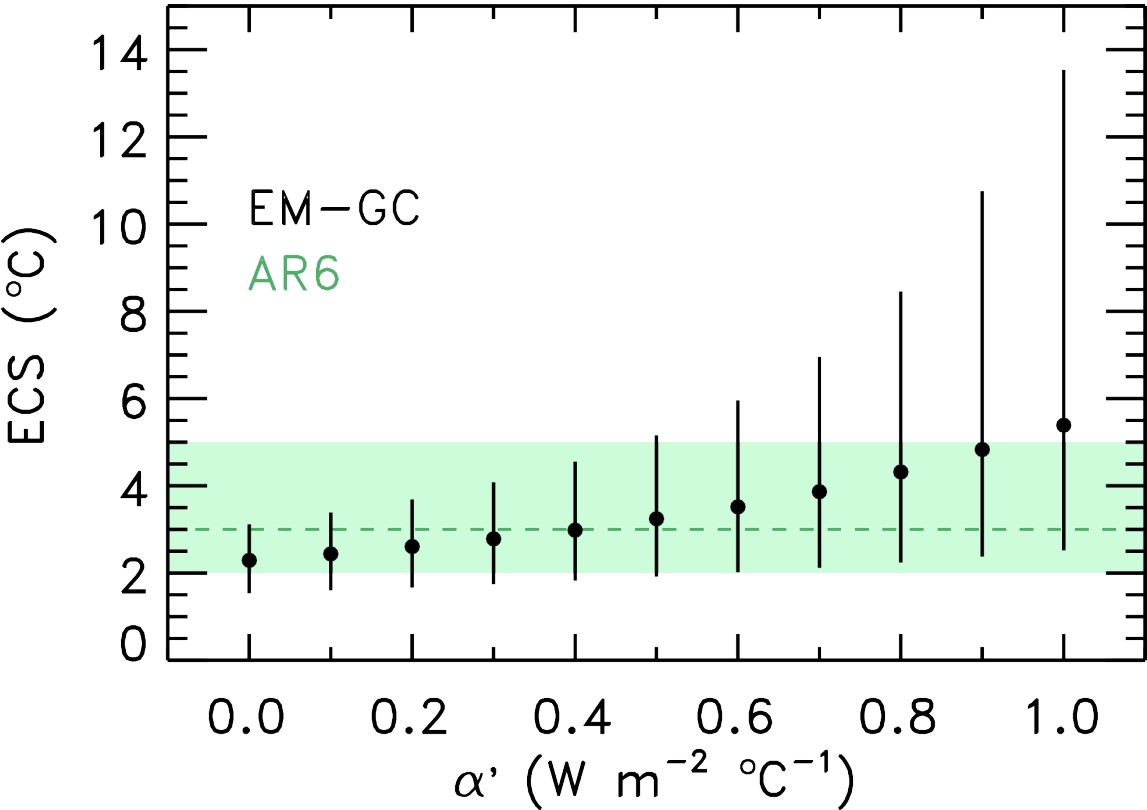

**Figure 5: Equilibrium Climate Sensitivity (ECS) as the function of the pattern effect (α', see text). Black vertical bars and circles correspond to the EM−GC 5−95% range, and 50% probability, respectively. The green shaded area and horizontal dashed line represent the AR6 very likely range, and central estimate of ECS respectively, from Table 7.13 of Forster et al. (2021). All results shown in this figure are based on inputs from the AR6 ERF Framework.**

For α' = 0 W m$^{-2}$ K$^{-1}$ (corresponding to no pattern effect), EffCS is equal to ECS and we obtain a median and range for ECS of 2.29 [1.54 to 3.11 ºC]. Using the AR6 best estimate of α' = 0.5 W m$^{-2}$ K$^{-1}$, we find ECS to be 3.24 [1.92 to 5.15 ºC], while the AR6 upper limit of α' = 1.0 W m$^{-2}$ K$^{-1}$ yields a median and 5−95% range of 5.39 [2.52 to 13.54 ºC]. Our estimate of ECS agrees extremely well with the AR6 central estimate and likely range of ECS, for the AR6 best-estimate of α', which equals 0.5 W m$^{-2}$ K$^{-1}$. This conclusion is consistent with Skeie *et al.* (2024b), who also found their estimate of climate sensitivity to be "almost identical" to the AR6 central value and very likely range of 3.0 [2.0 ºC to 5.0 ºC], upon using the AR6 best estimate of α'. Our range of ECS of [1.92 to 5.15 ºC] at α' = 0.5 W m$^{-2}$ K$^{-1}$ is also consistent with the recent assessment of Cooper et al. (2024), who found ECS to be between 1.4 ºC to 5.0 ºC based on analysis of data acquired during the Last Glacial Maximum (LGM). Our median estimate of 3.24 ºC for α' = 0.5 W m$^{-2}$ K$^{-1}$ is, however, higher than the LGM−based best estimate of 2.4 ºC. However, when combining the LGM−based assessment with other lines of evidence, Cooper et al. (2024) finds ECS to be 2.9 ºC [2.1 to 4.1 ºC], which agrees well with our estimate.

Additionally, Fig. 5 shows that the 5$^{th}$ percentile of ECS varies between about 1.5 to 2.5 ºC, depending on the magnitude of α', consistent with the AR6 assessment of ECS being greater than 1.5 ºC at a virtually certain level of confidence (Forster



et al., 2021). The 50[th] percentile estimates for ECS also all fall into the range of ECS assessed by AR6. Conversely, our 95[th] percentile estimates for ECS vary greatly with α', and carry a substantially larger level of uncertainty than the 5[th] and 50[th] percentile estimates. These findings are consistent with Chapter 7 of AR6, which stated "warming over the instrumental record provides robust constraints on the lower end of the ECS range (high confidence), but owing to the possibility of future feedback changes it does not, on its own, constrain the upper end of the range, in contrast to what was reported in AR5."

We now discuss how the rate of warming in recent decades relates to EffCS/ECS and the spatial pattern of global warming. Armour *et al.* (2024) found the central estimate and 5−95% range for the rate of warming between 1981 and 2014 to be 0.18 [0.15 to 0.21 ºC decade$^{-1}$], which corresponds to ECS of 2.7 [1.5 to 3.9 ºC] and EffCS of 2.3 [1.9 to 2.7 ºC]. The rate of warming obtained by Armour *et al.* (2024) is in close agreement with our AAWR estimate of 0.18 [0.13 to 0.21 ºC decade$^{-1}$] between 1974 and 2014. Furthermore, our EffCS estimate of 2.29 [1.54 to 3.11 ºC] closely matches that of Armour *et al.* (2024), albeit with a wider 5−95% range. Finally, our ECS estimate was found to be 3.24 [1.92 to 5.15 ºC] for α' = 0.5 W m$^{-2}$ K$^{-1}$, which is broadly consistent with Armour *et al.* (2024) value. Samset *et al.* (2023) found that CMIP6 models with ECS greater than about 3.0 ºC tend to provide an overly high estimate of the trend of observed warming in recent decades. Armour *et al.* (2024) suggest the overly high estimates of recent warming trends are related to CMIP5/6 historical simulations failing to reproduce observed SST patterns, particularly a cooling of the eastern tropical Pacific and a warming of the western Pacific. Weaver *et al.* (2024) reached the same conclusion based on analysis of top of the atmosphere albedo of clouds and aerosols from 340 nm radiances observed by NASA and NOAA satellite instruments. Armour et al. (2024) found that CMIP5/6 models with high ECS can nonetheless provide estimates of warming trends consistent with observations, after correcting for the biases in SST trend patterns. Weaver *et al.* (2024) showed that a significant number of CMIP6 models are unable to simulate the observed latitudinal pattern of cloud albedo trends over the Pacific Ocean, again tied to the observed cooling trend in the eastern tropical Pacific.

Our multiple linear regression – energy balance RCM, despite not considering the spatial pattern of warming, is able to reproduce estimates of EffCS and ECS consistent with the Armour *et al.* (2024) study, who explicitly consider the spatial pattern of global warming. We find that the range of observed warming in recent decades is consistent with the AR6 estimate of ECS and that the CMIP5/6 models that estimate a more rapid rate of warming may indeed fall into the "hot model" category (Tokarska et al., 2020b; McBride et al., 2021; Hausfather et al., 2022), potentially due to an overestimation of historical EffCS (Armour et al., 2024).

## 3.2 Probabilistic forecast on future warming

Here we quantify the magnitude of future warming based on projections of ERF for four SSP scenarios. Here, we show projections of GMST for only constant feedback because the GMST anomaly can be fit quite well for a wide-range of time invariant feedback, albeit spanning a large range of values, particularly for the Baseline Framework (Fig. 3c-d). We compare forecasts of GMST using the AR6 Framework with forecasts found using the Baseline Framework for the same SSPs, where "Baseline" refers to the state-of-knowledge prior to the release of AR6. Similarly to our analysis of AAWR and EffCS in





Section 3.1, forecasts of the GMST anomaly for the $\lambda_\Sigma - ERF_{AER,t}$ grid, weighted by an asymmetrical Gaussian function (Fig. 3a-b) that describes the AR5-based (Baseline) or AR6-based assessments of the likelihood of how much global warming by GHGs has been offset by tropospheric aerosols, is central to these probabilistic forecasts.

Figure 6 shows computed values of the GMST anomaly in the year 2100 ($\Delta T_{2100}$) for the AR6 Framework, for four SSPs. The colors correspond to values of $\Delta T_{2100}$ for model runs that provide a good fit to the observed GMST anomaly over the

1850−2019 training period, as defined by all three $\chi^2$ metrics described in Section 2.7 yielding values less than or equal to 2.

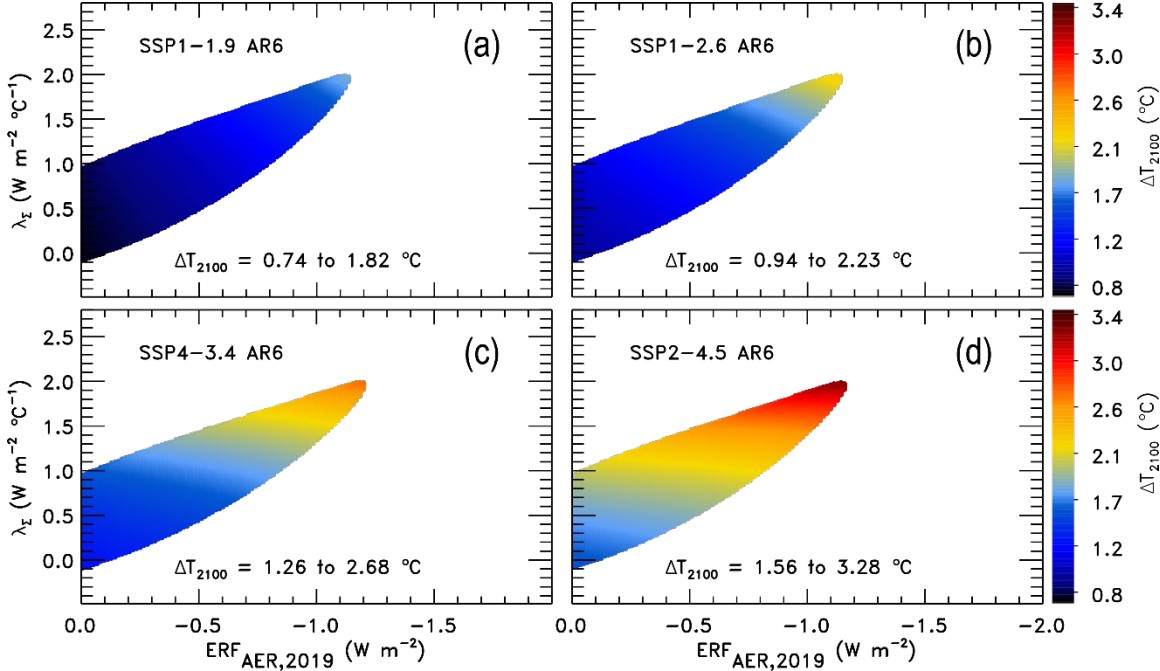

**Figure 6: GMST anomaly at the end of the century ($\Delta T_{2100}$) relative to a pre−industrial baseline for combinations of $\lambda_\Sigma$ and $ERF_{AER,2019}$. Colors denote values of $\Delta T_{2100}$ as indicated by the color bars on the right. Plots generated from EM-GC simulations trained on the HadCRUT5 GMST record, and use the AR6 ERF Framework. Only those combinations are colored where a good fit**

**on the HadCRUT5 GMST record was found. (a) $\Delta T_{2100}$ for SSP1−1.9. (b) $\Delta T_{2100}$ for SSP1−2.6. (c) $\Delta T_{2100}$ for SSP4−3.4. (d) $\Delta T_{2100}$ for SSP2−4.5. Values of $\Delta T_{2100}$ for the $\lambda_\Sigma - ERF_{AER}$ grid within the Baseline Framework are shown in Fig. S2.**

Table 1 details numerical values for $\Delta T_{2100}$ projections for the four SSPs. As described in Sect. 3.1, within the Baseline Framework the historical GMST record can be fit with combinations of high $\lambda_\Sigma$ and strong aerosol cooling. This difference results in a small shift in the upper boundary of $\Delta T_{2100}$, with values associated with the Baseline Framework being larger than

those for the AR6 Framework. Even though the climate record can be fit with much higher values of $\lambda_\Sigma$ within the Baseline Framework, ERF projections within the AR6 Framework are much higher than in the Baseline Framework (Fig. 1i), which offsets some of the $\lambda_\Sigma$ – driven difference. For SSP2−4.5, the upper estimate of $\Delta T_{2100}$ being 3.47 ºC corresponds to a value of $\lambda_\Sigma$ of about 2.5 W m$^{-2}$ ºC$^{-1}$ (Fig. S2) within the Baseline Framework. The upper boundary of $\Delta T_{2100} = 3.28$ ºC is found at a smaller climate feedback of about $\lambda_\Sigma = 2.0$ W m$^{-2}$ ºC$^{-1}$ for the AR6 Framework.



**Table 1: $\Delta T_{2100}$ for the four SSP scenarios studied. The first four rows correspond to data for the entire $\lambda_\Sigma - ERF_{AER,t}$ grid as shown in Figs. 6 and S2. The last four rows are associated with the EM−GC probabilistic assessment, with values derived from PDFs in Fig. 8 shown using the format of Median [5−95% range].**

| | SSP Scenario | Baseline Framework (°C) | AR6 Framework (°C) |
|---|---|---|---|
| Full $\lambda_\Sigma - ERF_{AER,t}$ grid | SSP1−1.9 | 0.73 to 2.06 | 0.74 to 1.82 |
| | SSP1−2.6 | 0.93 to 2.58 | 0.94 to 2.23 |
| | SSP4−3.4 | 1.15 to 3.01 | 1.26 to 2.68 |
| | SSP2−4.5 | 1.42 to 3.47 | 1.56 to 3.28 |
| Median [5−95% range] | SSP1−1.9 | 1.14 [0.81 to 1.87] | 1.34 [0.93 to 1.72] |
| | SSP1−2.6 | 1.46 [1.05 to 2.28] | 1.67 [1.18 to 2.13] |
| | SSP4−3.4 | 1.80 [1.31 to 2.77] | 2.10 [1.54 to 2.62] |
| | SSP2−4.5 | 2.18 [1.62 to 3.14] | 2.60 [1.92 to 3.20] |

Figure 7 shows our probabilistic forecasts of GMST in a time dependent fashion for the AR6 Framework. Time-dependent
projections of GMST for the Baseline Framework are in Fig. S3. Similar to AAWR and EffCS discussed above, the probabilistic forecasts of the GMST anomaly are provided by weighting model results that provide good fits to the climate record within the $\lambda_\Sigma - ERF_{AER}$ grid. The weighting procedure uses the Gaussian functions that describe the assessed likelihood of various values of RF of climate due to aerosols shown on the top panels of Fig. 3. The colors on Fig. 7 correspond to the probability of the $\Delta T$ anomaly being equal to or greater than a given value. The figure also shows the HadCRUT5 GMST
observations in black, as well as the likely range of warming for 2016 to 2035 provided by the authors of Chapter 11 of the AR5 report (Kirtman et al., 2013) in recognition of the fact that the CMIP5 models tended to over-estimate observed warming. Gold horizontal lines represent the 1.5 °C and 2.0 °C GMST anomalies relative to pre−industrial, while gold circles correspond to the years where the 1.5 °C and 2 °C thresholds are crossed with 5, 50 and 95% probability (hereafter termed crossover years). The probabilistic GMST anomaly forecast using SSP1−1.9 in the AR6 Framework lies below the 2 °C threshold until the end
of century throughout the ensemble (Fig. 7a), whereas the projection of GMST for SSP2−4.5 exceeds the 1.5 °C threshold in all cases (Fig. 7d).




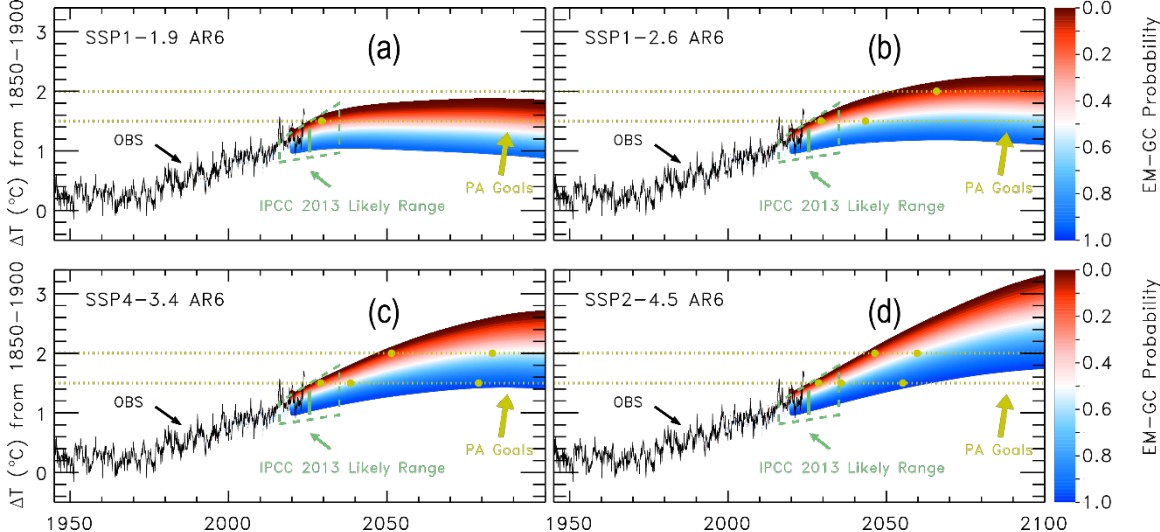

**Figure 7: Time-dependent probabilistic forecasts of the GMST anomaly beyond 2019 within the AR6 Framework. The black line represents the HadCRUT5 GMST record on which EM−GC was trained for these simulations, relative to an 1850−1900 baseline. Colors represent the probability of reaching a certain temperature or higher at a given time, as indicated by the color bars on the right. The green trapezoid represents the likely range of warming as shown in Fig. 11.25b of the IPCC AR5 report (Kirtman et al., 2013). The target and upper limit of the Paris Agreement are shown with horizontal lines in gold. Circle markers on these lines correspond to the projected GMST anomaly crossing these thresholds with the probability indicated by the colors. (a) GMST projections for SSP1−1.9. (b) GMST projections for SSP1−2.6. (c) GMST projections for SSP4−3.4. (d) GMST projections for SSP2−4.5. Results for the Baseline Framework are shown in the same fashion in Fig. S3.**

Table 2 provides the year that temperature anomaly thresholds are crossed for the median as well as the 5th and 95th percentiles of the probabilistic forecasts of ΔT. Figure 8 shows the probability distribution functions (PDFs) of $\Delta T_{2100}$. Results for both the Baseline and AR6 Framework are shown in Table 2 and Fig. 8, although we generally discuss only the results found for the latter. For the 5th percentile of the AR6 Framework, the 1.5 ºC threshold is crossed in either 2028 or 2029 for all four SSP scenarios. For the median member of our probabilistic forecasts, the 1.5 ºC threshold is crossed in the years 2035, 2038, and 2043 for the SSP2−4.5, SSP4−3.4, and SSP1−2.6 scenarios, respectively, whereas the 1.5 ºC threshold is not crossed for the median member of the SSP1−1.9 scenario. For the 95th percentile of the forecasts within the AR6 Framework, the 1.5 ºC warming threshold is crossed in 2055 and 2079 for the SSP2−4.5 and SSP4−3.4 scenarios, respectively, and is not crossed for the other two scenarios. Here, the 5th percentile has more warming than the 95th percentile, due to the fact that Fig. 7 shows probabilistic forecasts that ΔT will exceed a given amount.

Median projections of $\Delta T_{2100}$ within the AR6 Framework are found to be between 0.2 ºC (SSP1−1.9, SSP1−2.6) and 0.4 ºC (SSP2−4.5) greater than in the case of the Baseline Framework, mainly due to the larger projected anthropogenic ERF for AR6 (Fig. 1i). The PDF of $\Delta T_{2100}$ for the AR6 Framework is narrower and shifted a bit to the right (higher chance of more warming), compared to the Baseline Framework (Fig. 8), primarily due to two factors: the ability to obtain good fits to the GMST anomaly at the highest values of climate feedback only for the Baseline Framework, and the larger weight given to ensemble members with large, contemporary aerosol cooling in the AR6 Framework (Fig. 3).



Our median, 5<sup>th</sup> and 95<sup>th</sup> percentile estimates of $\Delta T_{2100}$ (Table 1) agree well with the long-term warming assessed by AR6, for years 2081−2100. Table 4.5 of AR6, which uses Global Surface Air Temperatures (GSAT) rather than GMST, projects warmings (median and 5−95% range) of 1.4 [1.0 to 1.8 ºC], 1.8 [1.3 to 2.4 ºC], and 2.7 [2.1 to 3.5 ºC] for SSP1−1.9, SSP1−2.6 and SSP2−4.5, respectively. As described in Chapters 1, 2 and 4 of AR6 (Chen et al., 2021; Gulev et al., 2021; Lee et al., 2021), GMST and GSAT differ by up to 10%, but the sign of the difference is highly uncertain. Our projections of end-of-century warming and for crossing the 1.5 ºC and 2.0 ºC GMST thresholds are generally consistent with the GSAT-based assessments of AR6 (Lee et al., 2021).

Finally, our study relies on SSP−based ERF data and does not factor in the significant decline of maritime sulfur emissions (and connected change in $ERF_{AER}$) caused by updated regulations on the sulfur content of ship fuels, introduced by the International Maritime Organization (IMO) in 2020. Varying estimates exist on the magnitude of $\Delta ERF_{AER}$ from the new IMO regulations and their impact on GMST/GSAT (Yuan et al., 2024; Skeie et al., 2024a; Quaglia and Visioni, 2024; Yoshioka et al., 2024). The reduction in shipping emissions may have brought forward global warming by 2−3 years (Gettelman et al., 2024; Jordan and Henry, 2024). Such a change would also help explain the unexpectedly high GMST anomalies of the year 2023, which have not otherwise been captured by climate model projections (Schmidt, 2024). The impact of IMO regulations on GMST in our model framework will be quantified in a future study.

**Table 2: Years of crossing the 1.5 ºC and 2.0 ºC GMST anomaly thresholds for the four SSP scenarios studied. For each entry, we present the 50% probability as our central estimate, as well as the 5−95% range. The label "n.c" is used consistent with Table 4.5 of AR6 (Lee et al., 2021) and corresponds to a given threshold not being crossed in the 2020−2100 period.**

| | 1.5 ºC Crossover | | 2.0 ºC Crossover | |
|---|---|---|---|---|
| | **Baseline** | **AR6** | **Baseline** | **AR6** |
| **SSP1−1.9** | n.c [2028 to n.c] | n.c [2029 to n.c] | n.c [n.c to n.c] | n.c [n.c to n.c] |
| **SSP1−2.6** | n.c [2029 to n.c] | 2043 [2029 to n.c] | n.c [2059 to n.c] | n.c [2065 to n.c] |
| **SSP4−3.4** | 2048 [2030 to n.c] | 2038 [2029 to 2079] | n.c [2050 to n.c] | 2083 [2051 to n.c] |
| **SSP2−4.5** | 2042 [2030 to 2080] | 2035 [2028 to 2055] | 2082 [2047 to n.c] | 2059 [2046 to n.c] |



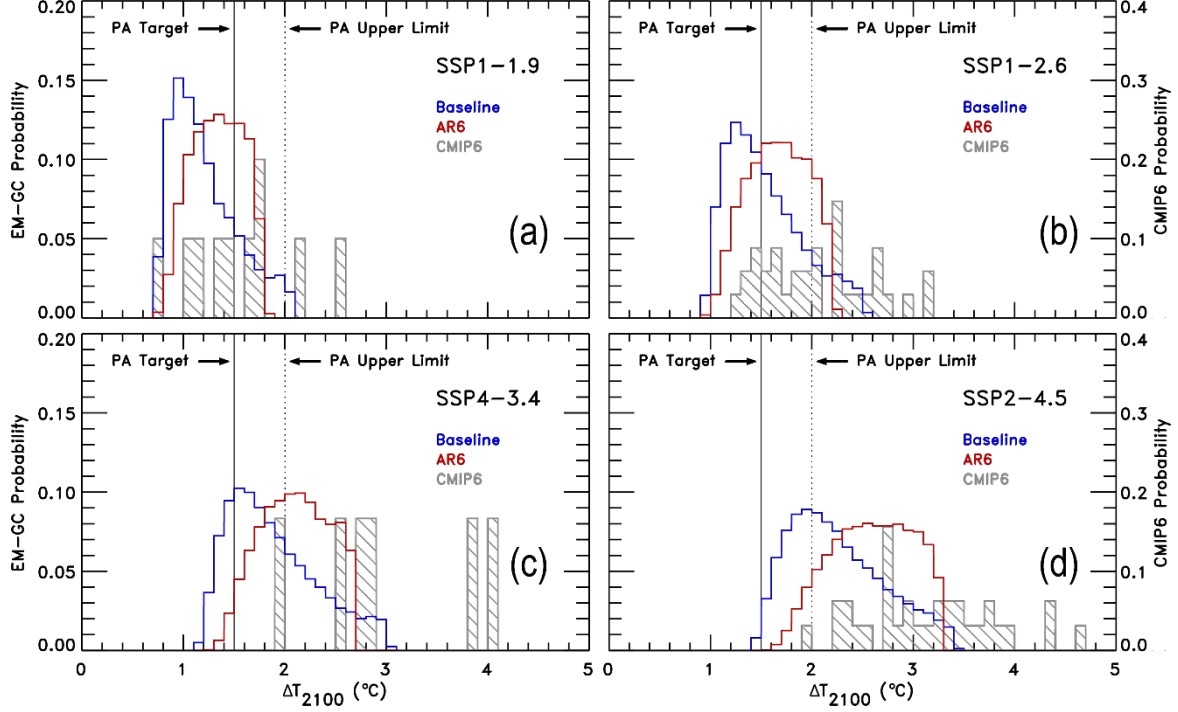

**Figure 8: Probability Distribution Functions (PDFs) for ΔT$_{2100}$ obtained from EM−GC simulations trained on the HadCRUT5 temperature dataset. Model runs for the Baseline and AR6 Frameworks are shown in blue and red, respectively. Grey color represents the PDFs obtained from a CMIP6 multi−model ensemble as described in Section 3.3.1 of McBride et al. (2021), and are shown for comparison with EM−GC results. The left-hand y axis corresponds to the EM−GC probabilities, and the right-hand y axis is for the CMIP6 probabilities. The PA target and upper limit are shown as solid and dashed vertical lines, respectively. (a) PDFs for SSP1−1.9. (b) PDFs for SSP1−2.6. (c) PDFs for SSP4−3.4. (d) PDFs for SSP2−4.5.**

Figure 8 also shows PDFs of ΔT$_{2100}$ from CMIP6 models. Consistent with many prior studies, including the Reduced Complexity Model Intercomparison Project (Nicholls et al., 2020; Nicholls et al., 2021), the free-running CMIP6 models exhibit larger end-of-century warming than projected by our model.

Tokarska *et al.* (2020b) reported that observationally constrained CMIP6 projections of end-of century warming are 9% to 13% lower than unconstrained CMIP6 projections for SSP1−2.6 and SSP2−4.5, respectively. Tokarska *et al.* (2020b) found the median and 5−95% ranges of end-of century warming relative to a 1995−2014 baseline to be 0.94 [0.41 to 1.46] °C and 1.84 [1.15 to 2.52] °C for SSP1−2.6 and SSP2−4.5, respectively, using observationally constrained CMIP6 models. Our values of ΔT$_{2100}$ in Table 1, relative to 1995−2014, are 0.81 [0.32 to 1.27] °C and 1.74 [1.06 to 2.34] °C for SSP1−2.6 and SSP2−4.5. Consequently, our quantification of ΔT$_{2100}$ is in good agreement with the findings of Tokarska *et al.* (2020b). As for the comparison with Fig. 4.2a of AR6 (Lee et al., 2021), it is important to highlight that our estimates use GMST, whereas Tokarska *et al.* (2020b) presents (adjusted) GSAT. Therefore, some of the differences between the forecasts can be explained by the difference in GSAT and GMST. Further, Chylek *et al.* (2024) recently found GSAT to be 2.41 °C in the year 2100 for SSP2−4.5 using a set of CMIP6 models that accurately reproduce the 2014−2023 warming, about 0.5 °C smaller than the average obtained from an unconstrained ensemble, in reasonably good agreement with our median estimate for ΔT$_{2100}$ of 2.6 °C.



In summary, our projections of $\Delta T_{2100}$ using the EM−GC model, trained within the AR6 Framework, produces estimates consistent with the output of observationally constrained CMIP6 models presented in numerous other studies. These estimates were obtained assuming climate feedback is constant between 1850 and 2100 (that is, α' = 0 W m$^{-2}$ K$^{-1}$). The close consistency between our $\Delta T_{2100}$ projections and the prior literature provides confidence in the virtue of the assumption that climate feedback has been mainly constant over the past century and a half, even though the actual values of climate feedback is not well known. The interested reader is directed towards Sect. 3.3.6 of McBride *et al.* (2021), for projections of $\Delta T$ in our model framework that allow the value of climate feedback to vary over time.

**Table 3: EM−GC computed probabilities of achieving the Paris Agreement target (1.5 ºC) and upper limit (2.0 ºC). Columns with the "Baseline" header represent EM−GC simulations using the Baseline Framework, while "AR6" represents simulations utilizing the AR6 Framework. The values presented in this table are derived from the PDFs shown in Fig. 8.**

|  | Paris 1.5 ºC | | Paris 2.0 ºC | |
|---|---|---|---|---|
|  | **Baseline** | **AR6** | **Baseline** | **AR6** |
| **SSP1−1.9** | 81% | 70% | 98% | 100% |
| **SSP1−2.6** | 54% | 32% | 87% | 85% |
| **SSP4−3.4** | 21% | 3% | 65% | 40% |
| **SSP2−4.5** | 1% | 0% | 35% | 8% |

We conclude this section by evaluating the probability of achieving both the target (1.5 ºC) and upper limit (2.0 ºC) of the Paris Agreement (PA). Table 3 provides the probability that end-of century warming will be below either the target or the upper limit, relative to pre-industrial conditions. These estimates were obtained from our probabilistic forecasts of $\Delta T$, for both the Baseline and AR6 Frameworks (Fig. 8). As shown above, median projections of $\Delta T_{2100}$ are larger within the AR6 Framework relative to the Baseline for all four SSPs, which leads to a decline in the probability of accomplishing the PA within the AR6 Framework (Table 3). For the SSP1−1.9 and SSP1−2.6 scenarios, the probability of limiting global warming to 2.0 ºC is high (at least 85%) for both model frameworks. For SSP4−3.4, the probability of limiting warming to 2.0 ºC falls from 65% (Baseline) to 40% (AR6). Most notably, the 2.0 ºC probability drops from 35% to 8%, for the SSP2−4.5 scenario. The 1.5 ºC warming probabilities for the AR6 Framework are all uniformly lower than for the Baseline Framework, with the SSP1−2.6 scenario dropping from 54% (Baseline) to 32% (AR6). The takeaway message from Table 3 is that, for society to have high confidence in achieving at least the upper limit of the PA, the radiative forcing of climate due to GHGs must be placed close to the SSP1−2.6 pathway over the coming decades. More aggressive reductions in GHG radiative forcing are needed to achieve the target of the PA, such as those of the SSP1−1.9 scenario.



## 4 Conclusions

The extent of global warming is proportional to the ERF from greenhouse gases and tropospheric aerosols. In this work, we use a multiple linear regression energy balance model (EM−GC) to quantify how updates to the ERF formulations of GHGs and tropospheric aerosols adapted by the IPCC AR6 report (termed AR6 Framework in our study) impact estimates of climate

sensitivity and projected future warming compared to forecasts made based on pre−AR6 datasets (Baseline Framework). Our study focuses on four policy-relevant SSP scenarios: SSP1−1.9, SSP1−2.6, SSP4−3.4 and SSP2−4.5 (O'Neill et al., 2014; O'Neill et al., 2016). Our model framework has two key elements: numerical representation of ocean heat export as well as the impact of uncertainty in radiative forcing due to tropospheric aerosols. The numerical values given below are results based on 160,000 member ensembles for each SSP.

First, we compare ERF for GHGs and tropospheric aerosols between the two Frameworks. We find that the projected ERF due to GHGs – particularly $CO_2$ and $CH_4$ − is considerably higher in the AR6 Framework relative to the Baseline, for all four of the SSP scenarios. This increase in $ERF_{GHG}$ originates from the updated assessments of future concentrations as well as updates to the formulations of ERF due to GHGs in AR6, in combination with updated adjustments to the RF of GHGs that now include responses throughout the troposphere, in addition to the stratosphere (Smith et al., 2018b; Hodnebrog et al., 2020;

Meinshausen et al., 2020; Forster et al., 2021; Smith et al., 2021a; IPCC, 2021b). End-of-century ERF is found to be up to 0.7 W m$^{−2}$ higher within the AR6 Framework relative to the Baseline for SSP2−4.5. While some minor deviation of end-of-century ERF from the nameplate RF value of SSP scenarios is expected (Van Vuuren et al., 2014), the AR6 Framework results in a level of increase in ERF such that end-of-century values are considerably larger than the nameplate RF for all four SSP scenarios.

The magnitude of Effective Climate Sensitivity (EffCS) inferred from the historical GSMT record was found to be 2.29 [1.54 to 3.11 ºC, 5−95% range] within the AR6 Framework, and 2.26 [1.45 to 4.37 ºC] for the Baseline Framework. The median value of EffCS is nearly identical between the two Frameworks, and there is a much narrower range within the AR6 Framework. Equilibrium Climate Sensitivity (ECS) is estimated to be 3.24 [1.92 to 5.15 ºC] within the AR6 Framework, using the AR6 best estimate for the pattern effect (α' = 0.5 W m$^{−2}$ K$^{−1}$). This estimate of ECS is quite similar to the AR6 assessment

of 3.0 [2.0 to 5.0 ºC] given in Table 7.13 of Forster *et al.* (2021). Overall, our estimates of EffCS and ECS within the AR6 Framework compare very well with values reported by several other recent studies (Armour et al., 2024; Cooper et al., 2024; Skeie et al., 2024b). The rate of human−induced warming between 1974 and 2014 (AAWR) was found to be 0.18 [0.13 to 0.21 ºC decade$^{−1}$] within the AR6 Framework, a slight increase relative to the central estimate and range of 0.16 [0.12 to 0.20 ºC decade$^{−1}$] for the Baseline Framework. Our estimate of AAWR within the AR6 Framework is also consistent with other recent

studies (Gulev et al., 2021; Forster et al., 2023; Samset et al., 2023).

Our projections of GMST show good consistency with GSAT estimates from observationally constrained CMIP6 ensembles but fall below the estimates of global warming when considering unconstrained CMIP6 ensembles. Overall, our estimates of EffCS, ECS, AAWR and end-of century warming using the AR6 Framework are all consistent with values found



by CMIP6 models that accurately reproduce observed warming trends over the past three or four decades. Our work highlights
the importance of ensuring that CMIP6 models used for policy purposes succeed in reproducing observed trends in GMST.

Our probabilistic projections on future warming consider the uncertainty in the magnitude of ERF from tropospheric aerosols, as assessed by AR6 and AR5, for the AR6 and Baseline Frameworks, respectively. Global warming projections that utilize the uncertainty in aerosol ERF is a common product of RCMs (Nicholls et al., 2020; Nicholls et al., 2021; Hope et al., 2017; McBride et al., 2021) and is not usually found using GCMs due to the computational demand of global models. Median
projections on end-of century warming using our RCM, are higher by 0.2 ºC to 0.4 ºC within the AR6 Framework relative to projections found in the Baseline Framework. This increase corresponds to a general decline in the likelihood of limiting warming to the Paris Agreement (PA) thresholds, for the AR6 Framework compared to the Baseline. Notably, model simulations using SSP2−4.5 and SSP1−2.6 − which reflect trends in the absence of further climate policies and full implementation of currently proposed emission targets (Meinshausen et al., 2024) – are found to offer 32% and 0% chance of
accomplishing the PA target of limiting the rise in GMST to 1.5 ºC by 2100. Model simulations conducted using the SSP4−3.4 (intermediate scenario between SSP1−2.6 and SSP2−4.5) and SSP1−1.9 (aggressive future reductions in GHG emissions) show probabilities of 3% and 70% of limiting warming to 1.5 ºC by end-of-century. The likelihood of limiting warming to the PA upper limit of 2.0 ºC is found to be 40% and 8% within the AR6 Framework for SSP4−3.4 and SSP2−4.5, respectively. There is high confidence (85% and 100% probabilities) that global warming could be limited to 2.0 ºC if the radiative forcing
of climate due to GHGs could be placed on either the SSP1−2.6 or SSP1−1.9 pathways.

## 5 Data availability

All data used as inputs of EM−GC are available from online resources. We have provided the links to these datasets below. The compiled input files used by EM−GC are also provided on Zenodo.org at 10.5281/zenodo.14720490 (Farago et al., 2025). The EM−GC output data is also provided in this Zenodo repository.

- SSP database (Baseline Framework): https://tntcat.iiasa.ac.at/SspDb/
   - Tropospheric $O_3$ RF (Baseline Framework): https://www.pik-potsdam.de/~mmalte/rcps/
   - AR6 Radiative Forcing (AR6 Framework): https://doi.org/10.5281/zenodo.5705391
   - MEIv2 and MEI.ext: https://psl.noaa.gov/enso/mei/ and https://psl.noaa.gov/enso/mei.ext/
   - PDO: http://research.jisao.washington.edu/pdo/PDO.latest.txt
- COBE SST data used to construct the IOD time series is available at: https://psl.noaa.gov/data/gridded/data.cobe.html
   - GloSSAC SAOD: https://asdc.larc.nasa.gov/project/GloSSAC
   - TSI: https://lasp.colorado.edu/sorce/data/tsi-data/
   - OHC Records:
      - Balmaseda: https://www.cgd.ucar.edu/cas/catalog/ocean/oras4.html
- Carton: https://www2.atmos.umd.edu/~ocean/soda3_readme.htm



- o Cheng: http://www.ocean.iap.ac.cn/pages/dataService/dataService.html?navAnchor=dataService
- o Ishii: https://www.data.jma.go.jp/gmd/kaiyou/english/ohc/ohc_global_en.html
- o Levitus: https://www.ncei.noaa.gov/access/global-ocean-heat-content/

## 6 Author contribution

EF updated the EM−GC model, performed the model simulations, conducted the data analysis and wrote the first draft of the manuscript. LM, AH and TC developed earlier versions of the EM−GC model and assisted in the review and editing of the manuscript. BB led the compilation of the IOD and SAOD datasets, and participated in the review and editing of the manuscript. RS supervised the project and participated in the review and editing of the manuscript.

## 7 Competing interests

The authors declare that they have no conflict of interest.

## 8 Acknowledgements

We thank the World Climate Research Programme for coordinating and promoting CMIP6 through its Working Group on Coupled Modelling. We also thank the climate modeling groups participating in CMIP6 for producing and making their model results available, the Earth System Grid Federation (ESGF) for archiving the data and providing access. We thank those who
contributed to Chapter 7 and Annex III of AR6 for providing values of ERF and other quantities used in this manuscript. Finally, we appreciate the financial support of the NASA Climate Indicators and Data Products for Future National Climate Assessments program during the early phase of this research.

## 9 Financial support

This research was supported by the National Aeronautics and Space Administration (grant no. NNX16AG34G).

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
