# Peer review of "AR6 updates to RF by GHGs and aerosols lowers the probability of accomplishing the Paris Agreement compared to AR5 formulations"

_EGUsphere, 2025_

## Author Response (AR1)

We thank both reviewers for a comprehensive evaluation of the submitted paper. We have revised the paper in the manner noted in the original response, that is posted in the response to reviewer 2 of the discussion section of the original submission:

We thank both Reviewers for taking the time to carefully read our manuscript and provide valuable comments. We now understand the Reviewers' concerns regarding the length of the manuscript. if allowed to proceed, we propose to revise the manuscript by greatly reduce the length, as described below.

The revised manuscript would be built around four key figures:

- (New Figure 1): Comparison of AAWR and EffCS between the Baseline and AR6 Simulations (Original Fig. 3).
- (New Figure 2): Comparison of Equilibrium Climate Sensitivity (ECS) as the function of the pattern effect parameter α' (Original Fig. 5).
- (New Figure 3): Comparison of time-dependent probabilistic projections for the four SSP scenarios studied (Original Fig. 7).
- (New Figure 4): Comparison of Probability Distribution Functions (PDFs) for the end-ofcentury warming for the four SSP scenarios studied (Original Fig. 8).

The figures would be reduced to four, relative to the eight figures in the main body of the initial manuscript. Additionally, we propose moving Figs. 1 and 6 of the original manuscript to the supplement, and completely removing Figs. 2 and 4 of the original manuscript.

We also propose to revise and greatly shorten the text of the manuscript to better reflect the key results of our paper, given below, which we now realize was obscured by the length of the introductory material:

- Projected ERF provided in AR6 for the SSPs is much greater than in the prior SSP dataset, from
  the original (baseline) SSP database. This excess is due mainly to updates in the ERF
  formulations for CO2 and CH4. Further, for each SSP scenario, the projected ERF given by AR6
  at the end of this century significantly exceeds the target radiative forcing associated with
  each given SSP scenario (original Fig 1, to be moved to Supplement).
- Historical GMST, when fit with the AR6 ERF datasets, corresponds to a narrower range of EffCS of 2.29 °C [1.54 °C to 3.11 °C] relative to EffCS inferred from the Baseline simulations 2.26 °C [1.45 °C to 4.37 °C]. (New Fig. 1)
- We provide new estimates of ECS using various values of the pattern effect parameter  $\alpha$ ', and find a range for ECS of 3.24 [1.92 to 5.15  $^{\circ}$ C] for the AR6 best estimate of  $\alpha$ ' (New Fig. 2). This analysis is a notable advance relative to the McBride et al. (2021) paper.
- When the AR6 ERF datasets are used, the simulated GMST in the future is considerably higher than that for Baseline simulations, a direct consequence of the increase in projected ERF,

resulting in a less optimistic chance of achieving the 1.5C and 2.0C goal and upper limit of the Paris Agreement (New Fig. 4)

 Our AR6-based forecasts of GMST still provide a lower projected warming than is given by many of the CMIP6 ensemble members (New Fig. 4). We propose to update Fig. 7 of the original manuscript, such that we shall now include the minimum-, maximum and mean projections of time-dependent GMST projections from CMIP6, as our New Fig. 3.

Again, we sincerely appreciate both reviews and we hope we will be allowed to submit a revised, significantly shortened manuscript in response to these comments. The revised manuscript would rely heavily on the McBride et al. (2021) paper for our methodology, as suggested by both Reviewers.

Citation: https://doi.org/10.5194/egusphere-2025-342-AC1

Below, we provide response to the detailed comments of each reviewer.

**Reviewer 1:**

**Summary**

The authors apply an existing multiple linear regression model to decompose the relative contribution of internal and external forcing factors to global mean surface temperature (GMST) change over the 20th and 21st century. They compare the influence of different assumptions about effective radiative forcing from various constituents (but primarily tropospheric aerosols) between two generations of the CMIP protocol (AR5, referred to as Baseline, and AR6). The authors show that their MLR model reproduces the majority of features of the GMST response, and the effective climate sensitivity, simulated by a range of previous simplified and comprehensive modelling efforts. They use this information to provide probabilistic estimates of GMST remaining below Paris targets (1.5C and 2.0C).

**Major comments**

The paper is well researched, and well written; the figures are clear and communicate the main findings of the analysis. I believe that the conclusions reached are appropriate based on the methods and evidence presented. However, I cannot recommend publication of this manuscript for the following reasons.

First, this reviewer found that the authors have not adequately communicated what are the primary novel contributions of the research. On the contrary, in virtually all cases in the results sections, the authors highlight that their results are consistent with previous studies. This holds true for previous studies using simplified or intermediate-complexity models, and comprehensive modelling like CMIP6. This is a very well-studied field over the past decade, and the authors must articulate clearly how this research advances the discipline beyond what the myriad of previous studies has done.

We have revised the paper to focus, much more clearly, on the "primary novel contributions" of our research. The new abstract reads as follows:

We provide a reduced complexity climate model (RCM) evaluation of how the IPCC WG1 Sixth Assessment Report (AR6) updates to the time series of the future atmospheric concentrations of GHGs, the effective radiative forcing (ERF) of GHGs, and the ERF of tropospheric aerosols (ERFAER) affect attributable anthropogenic warming rate, climate sensitivity, and the likelihood of achieving either the goal (1.5 °C) or upper limit (2 °C) global warming thresholds of the Paris Agreement. This evaluation is conducted for four selected Shared Socioeconomic Pathway (SSP) scenarios: SSP1-1.9, SSP1-2.6, SSP4-3.4, and SSP2-4.5. Throughout, we compare and contrast these AR6 updates to the state of knowledge that existed prior to the publication of AR6, and provide probabilistic model simulations based on an evaluation of the impact in the uncertainty of ERFAER and climate feedback. Our most important findings are that the rate of human-induced warming between 1975 and 2014 is 0.18 [0.13 to 0.21] oC decade-1 within the AR6 framework (range reflects the 5th and 95th percentiles), which is considerably lower than values found by many Earth System Models (ESMs) that participated in Phase 6 of the Coupled Model Intercomparison Project (CMIP6). Effective Climate Sensitivity (EffCS) inferred from the historical Global Mean Surface Temperature (GSMT) record was found to be 2.29 [1.54 to 3.11] oC using the ERF datasets from AR6 as model inputs. Upon adoption of the AR6 best estimate for the pattern effect (that is, 0.5 W m-2 0C -1), we find values for Equilibrium Climate Sensitivity (ECS) of 3.24 [1.92 to 5.15] °C, which is guite similar to the AR6 assessment of 3.0 [2.0 to 5.0] °C for ECS. The hallmark of our RCM is the ability to conduct large (here, 160,000 member) ensemble forecasts of global warming. These calculations show that AR6 updates to the ERF of GHGs and aerosols result in a considerable decline in the likelihood of limiting warming to either 1.5 or 2 °C, compared to prior knowledge, for the same future emissions scenarios of GHGs. The likelihood of limiting global warming to 2.0 °C by end-of-century is found to be 100%, 85%, 40%, and 8%, for the SSP1-1.9, SSP1-2.6, SSP4-3.4, and SSP2-4.5 scenarios, respectively, based on the AR6 ERF datasets. Similarly, the ensembles run using the AR6 updates yield likelihoods of 70%, 32%, 3%, and 0% of limiting warming to 1.5 °C by end-ofcentury, for the same four SSPs.

Second, while the authors must be commended on their attention to detail and the depth of the research undertaken, the manuscript is much too long considering the paucity of new results being presented. At times it felt like a PhD thesis; for example, Section 2.3 provides many textbook-level definitions of ERF for various atmospheric constituents, and Section 2.5 describes every assumption for the inputs of the regression model in intricate detail. This raised the question of who exactly is the target audience for this work? Since the aim is to publish in ESD, there should be an assumption that interested readers will have sufficient background knowledge in climate change science to trust that emissions inventories, sources of natural/internal climate variability etc. are properly referenced and incorporated without the need for such a detailed assessment here. Potentially important and relevant previous literature was also excluded; for example, see:

We have taken this comment to heart in the revised version of the paper. We have tried our best to focus the text on the important, new aspects of the study in the revised manuscript.

The original submission lacked a citation to the Foster and Rahmstorf 2011 paper noted by the reviewer, as well as Lean and Rind (2008), Lean and Rind (2009), and Zhou and Tung (2013). We have added the following sentence to Section 2:

The MLR component of the model is responsible for quantifying the influence of various natural factors on the GMST in a manner similar to other MLR-based analyses of the climate system (Lean and Rind, 2008, 2009; Foster and Rahmstorf, 2011; Zhou and Tung, 2013)

and we have added the following sentence to Section 2.2.1:

Our method for the evaluation of AAWR is similar to earlier, MLR based studies (Lean and Rind, 2008, 2009; Foster and Rahmstorf, 2011; Zhou and Tung, 2013), except that we quantitatively account for the impact of the uncertainty in the RF of aerosols and the strength of climate feedback on the possible range of AAWR.

Our consideration of the uncertainties in the RF of tropospheric aerosols, the strength of climate feedback, as well as ocean heat export is a unique feature of our quantitative evaluation of the possible range in AAWR, which is considerably wider than most other evaluations.

Third, the linear modelling framework itself does not appear to be a new contribution (e.g., McBride et al. 2021). Therefore, it is somewhat surprising that potentially important limitations of the linear model approach are not investigated or advanced in this research. For example, on L513 the authors describe needing to include a third constraint (consistency with recent observed temperature trends) in order to yield solutions that match the GMST time series over the recent past. Given the importance of ERF\_CO2 and ERF\_AER on the GMST predictions from the EM-GC model, this suggests a probable role for nonlinear interactions between aerosol and CO2 forcing that the current model cannot capture. It would have been interesting to see this commented on, if not addressed, in the research.

The use of the chi-squared recent metric, which appeared on line 513 of the original submission, is a model element that was described in Section 2.1 of McBride et al. This metric is needed because the uncertainty associated with measurements of GMST is quite large.

The sentence in the original submission that had read:

The  $\chi^2_{RECENT}$  metric is used because without this constraint, some solutions with values of  $\chi^2_{ATM} \le 2$  have a visually poor simulation of the rise in GMST over the past 4 to 5 decades (McBride et al., 2021).

has been changed to read:

The  $\chi^2_{RECENT}$  metric is used because without this constraint, some solutions with values of  $\chi^2_{ATM} \le 2$  have a visually poor simulation of the rise in GMST over the past 4 to 5 decades, due to the large uncertainty associated with early measurements of  $\Delta T$  (McBride et al., 2021).

The most important non-linear component of our analysis is the possibility that climate feedback has changed over time. This possibility is examined, in great detail, within Section 3.3.6 of McBride et al. (2021). We have added the following sentence to Section 2.2.2 of the revised paper:

The projections of  $\Delta T$  shown in Section 3 assume that the climate feedback parameter,  $\lambda_{\Sigma}$ , is constant over time. Support for this assumption is given by the temporal invariance of the residual between measured and modeled values of  $\Delta T$ , over the past century and a half, as shown in Fig. 14 of McBride et al. (2021). If the true value of  $\lambda_{\Sigma}$  varies over time, as has been suggested based on analysis of CMIP5 (Marvel et al., 2018; Rugenstein et al., 2020) and CMIP6 (Dong et al., 2020; Salvi et al., 2023), then the analysis conducted by McBride et al. (2021) indicates that our end-of-century projections of global warming could be biased low by a few tenths of the degree Celsius. Regardless, the primary contributor to the uncertainty in end-of-century warming is the imprecise knowledge of ERFAER.

Fourth, one of the major findings of the research highlighted by the authors is the apparent increase in warming rate under the AR6 assumptions compared to pre-AR6 (baseline). However, on L502 the authors state that the 6\% higher climate sensitivity in AR6 comes from applying the published formula for ERF\_CO2 from AR6 that is larger than the pre-AR6 formula provided by Myhre et al. (1998). Therefore, it appears that the increased warming rate is "baked-in" to the EM-GC model, rather than an emergent property, making the findings of more warming and a lower probability of remaining below the Paris targets largely unsurprising.

The 6% larger value of ERF $_{CO2}$  in AR6 compared to Baseline has no substantial impact on our numerical evaluation of either AAWR or end-of-century warming, because both quantities depend on the time rate of change of radiative forcing. This 6% increase does not alter any of the slopes of the ERF terms.

The increased warming we find for the AR6 framework, relative to the Baseline framework, is a consequence of the larger rise in total anthropogenic ERF, ERFANTH, from the middle of the prior century to the end of this century. This change in slope is driven largely by AR6 versus AR5 assessments of aerosol cooling as well as AR6 updates to end-of-century atmospheric concentrations of CO2 and CH4.

We have added the following sentence to end of Section 2.1.3:

One final, important difference between the two frameworks is the steeper rise in ERFANTH between about 1960 and present within AR6 compared to Baseline, which is attributable to an assessed best value of much stronger aerosol cooling over the latter part of the prior century in AR6 relative to the Baseline (Fig. S1g).

and the following two sentences to Section 3.2:

Median projections of  $\Delta T_{2100}$  within the AR6 framework are about 0.2  $^{\circ}$ C (SSP1–1.9, SSP1–2.6), 0.3  $^{\circ}$ C (SSP4-3.4), and 0.4  $^{\circ}$ C (SSP2–4.5) greater than found using the Baseline framework. This difference originates from the fact that projected ERF at the end of the century is higher in the AR6 framework than in Baseline, for all four SSPs, which is driven by higher end-of-century atmospheric concentrations of CO2 and CH4 in AR6 (Fig. S1i).

to clarify these points.

**Minor comments:**

-L23: The overlap of the baseline and AR6 confidence intervals suggests that the statistical evidence for an increase in the mean is rather weak.

We agree. The comparison of the AR6 and Baseline framework values for AAWR is now omitted from the abstract. Rather, in the new abstract noted above, we focus on a comparison of our value of AAWR over the time period 1975 to 2014, to that inferred from free running ESMs over this same time period. The relevant new sentence in the new abstract is:

Our most important findings are that the rate of human-induced warming between 1975 and 2014 is 0.18 [0.13 to 0.21] oC decade-1 within the AR6 framework (range reflects the 5th and 95th percentiles), which is considerably lower than values found by many Earth System Models (ESMs) that participated in Phase 6 of the Coupled Model Intercomparison Project (CMIP6).

-L119 and L122: do the authors mean to say adopted, rather than adapted?

Thanks! The word "adapted" is not used in the revised paper.

-L152: Can the authors provide the proportions for the different effects in this attribution? Are CO2 concentrations the dominant effect?

The issue of higher ERF due to GHGs is now addressed in a meticulous fashion in Sections 2.1.1 (Atmospheric Concentrations of Greenhouse Gases) and Sections 2.1.2 (Radiative Forcing of Greenhouse Gases), as well as Fig. S1. It is challenging to rank  $CO_2$  or  $CH_4$  as being most important, since for some of the SSPs the change in ERF due to  $CO_2$  is larger than the change due to  $CH_4$ , and vice-versa for the other SSPs. So, we have written:

The middle row of Fig. S1 compares time series of ERF due to  $CO_2$ ,  $CH_4$ , and  $N_2O$ , for the Baseline (dotted lines) and AR6 (solid lines) frameworks. The results shown in this middle row reflect the AR6 updates to both the ERF and the future atmospheric abundances of GHGs. Values of ERF are higher in the AR6 framework compared to Baseline, with particularly large increases found for the ERFs of  $CO_2$  and  $CH_4$  for the SSP4–3.4 and SSP2–4.5 scenarios. Finally, Fig. S1h compares ERF due to all GHGs, for the Baseline and AR6 frameworks. The largest increase in ERF, among the four SSP scenarios considered, is found for SSP4–3.4 and

SSP2-4.5, with end-of-century increases of 0.6 and 1.0 W m-2, respectively. A similar qualitative conclusion was reached by Fredriksen et al. (2023), who contrasted projections of ERF from CMIP5 models with those from CMIP6 models, and found that CMIP6 models project higher levels of ERF by the end of the century relative to CMIP5 models.

-L205: Can the authors comment on why the ERF\_AER value changes by so much (15% larger) when the time period is shortened by only 5 years (ending in 2014)?

We have added the following text to explain the change of ERFAER in AR6, relative to AR5:

It is beyond the scope of this paper to delve deeply into the cause of the differences between the AR5 and AR6 estimates of ERFAER. It is somewhat surprising that the AR6 update to the best estimate of ERF AER in the year 2019 exhibits more cooling than the AR5 best estimate that reflected conditions out to 2011, because individual time series of ERFAER in both AR5 and AR6 (Fig. S1g) exhibit a considerable decline in the absolute value of ERFAER over the 2011 to 2019 period of time. This decline was driven by successful efforts to reduce the emissions of aerosol precursors, by various entities throughout the world, due to the public health concerns of aerosols (Smith and Bond, 2014; Fu et al., 2021). The primary reason for larger aerosol cooling in the AR6 best estimate of ERFAER, despite the 8-year extension in end year, is the nearly factor of two increase in the assessed value of cooling due to the aerosol indirect effect from AR5's best estimate of -0.45 [0.0 to -1.2] W m-2 to the AR6 best estimate of -0.84[-0.25 to -1.45] W m-2. A significant decline in the best estimate of black carbon warming in AR6  $(0.11 [-0.20 \text{ to } 0.42] \text{ W m}^{-2})$  compared to AR5  $(0.4 \text{ W m}^{-2} [0.05 \text{ to } 0.80] \text{ W m}^{-2})$  also contributes to the decline in the absolute value of ERFAER in AR6, compared to AR5. There are other updates in the AR6 approach for ERFAER, as summarized in Sect. 7.3.3 of Forster et al. (2021).

-L235: This paragraph is very unclear. What is the single best estimate ERF\_AER time series? What portion of the difference is highlighted?

This longer new paragraph, starting on line 172 of the revised paper, replaces the two sentence paragraph that indeed was unclear in the original submission:

Time series of ERFAER are vitally important inputs to our EM–GC. A hallmark of our approach is to span a wide range of possible time series of ERFAER, as well as a model parameter  $\lambda_{\Sigma}$  that represents the sum of all climate feedbacks, retaining for further analysis the members of this ensemble that satisfy three goodness-of-fit constraints, to the: 1) 170-year GMST record; 2) GMST record over the past 8 decades (formally, 1940 to 2019); 3) the ocean heat content record that begins in 1955. Further details of this ensemble approach are given in Sect. 2.1 of McBride et al. (2021). Figure S2 illustrates our approach for generating an ensemble of ERFAER time series for the SSP2–4.5 scenario, within the AR6 framework. The solid black line shows the AR6 assessed best value of the time series of ERFAER. An ensemble is created by scaling this time series by various constant multiplicative factors, with the color scheme chosen to highlight the numerical value of ERFAER in 2019. A similar approach is used for the Baseline

framework, relying upon time series of ERFAER obtained from the aforementioned PICR website, as detailed in Sect. 2.5 and Fig. S7 of McBride et al. (2021). While one can envision a more sophisticated approach that allows for the alteration of the shape of ERFAER, in addition to the magnitude, the actual ERFAER responds quickly to changes in precursor emissions due to the short lifetime of tropospheric aerosols. Generally, historical aerosol precursor emissions are fairly well known (e.g. Hoesly et al. (2018)). The more sophisticated approach of Smith and Bond (2014), which relied upon a RF parametrization tied to the emission of sulfate, black carbon, and organic carbon aerosols, resulted in an ensemble of time series for ERFAER that exhibit nearly the same shape, with quite different peak cooling.

-L690: Why did the authors elect to not examine a business-as-usual/ high emission scenario like SSP5-8.5?

As in McBride et al. (2021), we focus on the most prominent (that is, Tier 1 or Tier 2) SSPs that have the potential to allow society to meet the goal (1.5 °C warming) or upper limit (2 °C warming) of the Paris Agreement, plus the SSP2–4.5 scenario, which actually mirrors the current RF of climate by the three major GHGs most closely than other SSPs. The relevant text at the end of the Introduction reads as follows:

Here, we examine four policy-relevant SSP scenarios: SSP1–1.9, SSP1–2.6, SSP2–4.5, and SSP4–3.4 from Tier 1 and Tier 2 of the ScenarioMIP protocol (O'Neill et al., 2016). These were chosen because SSP2–4.5 is the SSP scenario most consistent with recent trends in the anthropogenic emissions of GHGs and aerosols (Meinshausen et al., 2024), while the other three SSPs we have chosen all offer more aggressive means for climate mitigation than the SSP2–4.5 scenario.

-L855: This conclusion is challenging, because the agreement between EM-GC outputs and the observed GMST timeseries is explicitly built in to the EM-GC model, whereas for the majority of CMIP6 models they are freely running through the 20th Century. Whether this reduces the value of those simulations is a matter for debate; perhaps a more nuanced view is that it affects the types of questions that one should ask of the CMIP-class models.

The sentence in question, in the original submission, read as follows:

Our work highlights the importance of ensuring that CMIP6 models used for policy purposes succeed in reproducing observed trends in GMST.

This sentence has been removed from the revised paper.

We do draw attention to the so-called "hot model" problem of ESMs in two places within section 3.2, where it is now stated:

Numerous studies have similarly concluded that many of the ESMs central to CMIP6 tend to provide estimates of the rate of global warming due to human activity (that is, AAWR) that exceeds empirically based estimates of AAWR (Tokarska et al., 2020b; Nijsse et al., 2020;

McBride et al., 2021, Chylek et al., 2024), which Hausfather et al. (2022) have termed the "hot model problem".

**as well as:**

Figure 4 shows the PDF of  $\Delta T_{2100}$  found with EM-GC for the four SSP scenarios, using the AR6 and Baseline frameworks. The height of the bars corresponds to the probability of  $\Delta T_{2100}$  being in the range defined by the width of each column. Figure 4 also shows PDFs derived from a CMIP6 ESM ensemble, as detailed by McBride et al., (2021). As expected, based on the "hot model problem" described above, our projections of  $\Delta T_{2100}$  within both the Baseline and AR6 frameworks fall on the lower end of the projections from the CMIP6 ensemble. Furthermore, the EM-GC based PDF for the AR6 framework tends to be shifted towards higher values of  $\Delta T_{2100}$  than found for Baseline, with a smaller tail, behaviors that are consistent with higher end of century RF of the climate within the AR6 framework (Fig. S1i), as well as the ability to fit the climate record with higher values of climate feedback (model parameter  $\lambda_{\Sigma}$ ) in the Baseline framework (Fig. 1).

**Reviewer 2:**

In this manuscript, the authors use the Empirical Model of Global Change (EM-GC) to explore how updates to effective radiative forcing, as reported in the IPCC Sixth Assessment Report (AR6), influence the following climate metrics: effective climate sensitivity, the rate of attributable anthropogenic warming, and projections of future warming. The manuscript concludes with an assessment of how these updates affect the likelihood of meeting the climate policy goals set by the 2015 Paris Agreement.

I was genuinely excited to see the EM-GC used in this context, and I believe this study has the potential to make a meaningful contribution to the literature. The research question is both timely and important, and I encourage the authors to continue developing this work. However, in its current form, the manuscript faces some structural challenges and clarity issues that make it difficult to fully appreciate the significance of the results. I would strongly encourage the authors to revise and resubmit, as I believe that with improvements, this paper could become a valuable addition to the field.

Much thanks for these kind words. Our revised document is much shorter, 567 lines, than the original submitted document, which had been 870 lines.

While I did not feel that a detailed, line-by-line review was appropriate at this stage, I would like to share a few broader comments that I hope will be helpful in guiding the revision.

At times, the manuscript reads like a blend of two distinct papers — part literature review, part research article. I recognize and appreciate the substantial effort the authors have put into the background material, and the breadth of the literature covered is impressive. That said, I feel that the extensive background somewhat overshadows the more novel and exciting aspects of the authors' analysis. I would recommend streamlining the background, particularly in the methods and data sections, and relying more heavily on citations to established work, which would allow the new contributions to stand out more clearly.

**We have fully followed this suggestion.**

I think referencing prior EM-GC literature more explicitly could help improve clarity in the model description. For example, Section 2.4 closely resembles McBride et al. (2021), and equations (1)–(4) appear to be the same as those in that manuscript. Could the authors clarify whether these equations are indeed unchanged, or if they have been modified in this study? Providing that clarification will help situate the current work within the existing EM-GC framework and highlight any new developments more effectively.

The equations are the same. As such, we have omitted the model equations from the revision. The original submission included 7 equation in Main, and 4 in Supplement. The revised paper has 2 equations in Main, and 3 in Supplement.

Another question that arose when reading this manuscript was how do the authors deal with the issue with the ERF of aerosols in the AR6 analysis (see Zelinka 2023). Did the authors account for or

correct for this basis their analysis? A comment of if/how the authors address this should be included in the manuscript or discussed as a potential limitation of the study.

We have decided to continue to base the analysis of ERF of aerosols on the AR6 time series, in part because the Zelinka et al. (2023) paper focuses mainly on the AR6 assessment of delta\_ERF between 1750 and 2000, and between 1750 and 2014. Much of our paper is based on the 1750 to 2019 time period central to many ERF estimates of AR6. We have added the following new paragraph to section 2.1.3 of the revised paper:

Recently, Zelinka et al. (2023) pointed out two coding errors in the Smith et al. (2020) paper that influenced the AR6 evaluation of ERFAER. These two errors largely cancel for the evaluation of ERFAER. The Zelinka et al. (2023) best estimate and standard deviation of ERFAER, over 1750 to 2014, is  $-1.09 \pm 0.24$  W m-2, which is slightly less aerosol cooling than the AR6 estimate of -1.3 [-0.6 to -2.0] W m-2 for the same time period. Given the "medium confidence" associated with the assessed value of ERFAER noted in Chapter 7 of AR6 (Forster et al., 2021), the lack of evaluation of ERFAER by Zelinka et al. (2023) for the 1750 to 2019 time period that is central to our study, and the focus within Zelinka et al. (2023) on the evaluation of the various components of ERFAER for contemporary periods of time rather than the historical evolution of ERFAER, we have decided to use the AR6 historical time series for aerosol cooling as presented in the assessment.

I appreciate the authors' effort and I am confident that with thoughtful revisions, this work can make a valuable impact. I look forward to seeing a future version of this manuscript and the contributions it will bring to our field.

Thanks for these kind words. We look forward to your comments on the revised manuscript.

---

## Author Response (AR2)

**Reviewer 1:**

Thank you for thoughtfully addressing my initial round of feedback, either within the manuscript text or through the author's notes where appropriate. It is evident that you have taken the feedback from both reviewers seriously and have invested significant effort into revising the manuscript, resulting in substantial improvements to its quality and clarity. Your dedication to enhancing the manuscript is greatly appreciated.

That said, there are still a few minor outstanding issues that should be addressed prior to publication. I am confident that resolving these will further strengthen the final version of your work. My comments below are grouped into the following categories, more general vs. specific line by line comments.

Thank you again for your hard work on this manuscript. I look forward to seeing the polished final version.

Thanks for these very kind words. We sincerely appreciate the constructive comments of this review.

**General Comments**

The repeated use of phrases such as "our EM-GC model" and "our model" struck me as somewhat unusual in the context of model description and application papers. Typically, authors in such papers refer to "the model," use the model's name directly, or specify the version to ensure clarity and consistency. While there is nothing technically incorrect about the phrasing used, it did make me wonder if the authors were intentionally aiming to differentiate their model from another existing EM-GC model. Clarifying whether this is the case or adopting more standard language might help avoid any potential confusion for readers.

We have reduced the use of the word "our" from 53 occurrences in the most recently reviewed version of the paper, to 25 occurrences in the revised manuscript.

The methods section has noticeably improved, and I appreciate the effort put into enhancing its detail and structure. However, as a reader, I still have some questions regarding certain aspects of the methodology. Addressing these points could significantly enhance the reproducibility and transparency of the overall manuscript.

Radiative Forcing Time Series: Since subsection 2.1.2 Radiative Forcing of Greenhouse Gases is part of 2.1 Model Inputs, does this mean that the radiative forcing time series are considered inputs to the EM-GC model? This seems to be implied by the statement: "The EM-GC relies on time series of the RF due to GHGs." If so, what role do the greenhouse gas (GHG) concentrations play within the model?

GHG concentrations are used as input for our calculation of the RF due to GHGs, which is then input to the model.

**The sentence that had read:**

The EM-GC relies on time series of the RF due to GHGs, computed from time series of the atmospheric abundance of each gas.

**has been changed to read:**

The EM-GC uses as input time series of the RF due to GHGs, computed from time series of the atmospheric concentration of each GHG.

**to address this comment.**

Role of HadCRUT5 GMST Anomaly: Could you clarify whether the HadCRUT5 global mean surface temperature (GMST) anomaly serves as an input to force the model simulation during the historical period? Alternatively, does the EM-GC model use historical GHG concentrations to drive the simulation from 1850 to 2014 and then use the HadCRUT5 GMST anomaly to constrain the model runs? Expanding on this distinction would help clarify the methodological framework.

We are sympathetic to this comment, as the model description is terse. The terse nature of the model description was a change made, in between submission of a much longer manuscript, and the revised version that was most recently reviewed. The two sentences noted below have been edited to address this comment.

**We have changed the sentence that had begun on line 196 that had read:**

The EM-GC output shown below relies entirely on simulations constrained to match the HadCRUT5 GMST anomaly ( $\Delta T$ ) record (Morice et al., 2021) over the years 1850–2019.

**to now read:**

The EM-GC output shown below relies entirely on simulations that we constrain to match the HadCRUT5 GMST anomaly ( $\Delta T$ ) record (Morice et al., 2021) over the years 1850–2019.

**Also, the sentence that began on line 172 that had read:**

A hallmark of this approach is to span a wide range of possible time series of ERFAER, as well as a model parameter  $\lambda_{\Sigma}$  that represents the sum of all climate feedbacks, retaining for further analysis the members of this ensemble that satisfy three goodness-of-fit constraints, to the: 1) 170-year GMST record; 2) GMST record over the past 8 decades (formally, 1940 to 2019); 3) the ocean heat content record that begins in 1955.

**has been changed to now read:**

A hallmark of this approach is to span a wide range of possible time series of

ERFAER, as well as a model parameter  $\lambda_{\Sigma}$  that represents the sum of all climate feedbacks, retaining for further analysis the members of this ensemble that satisfy three goodness-of-fit constraints, to the observed: 1) 170-year GMST record; 2) GMST record over the past 8 decades (formally, 1940 to 2019); 3) ocean heat content record that begins in 1955.

Use of AR5 Simulations: In section 3.2 Probabilistic Forecast of Future Warming, there is some discussion comparing results with those from AR5. Were any AR5 simulations specifically run with the EM–GC model? If so, it would be helpful for the authors to provide details about which AR5 simulations were conducted. For example, was the CMIP5 historical dataset used instead of the CMIP6 historical GHG concentrations from ScenarioMIP?

No, we did not run any AR5 simulations in this manuscript. AR5 relied on the Representative Concentration Pathway (RCP) scenarios, which are the basis of our earlier Mascioli et al. (2012) and Hope et al. (2017) studies. In the current manuscript, we make important reference to CMIP5 ESM output by the inclusion of the trapezoid placed on Figure 3, which originates from Figure 11.25b of AR5. This trapezoid represents the likely range of warming over the time period 2016 to 2035, provided by the authors of Chapter 11 of AR5, in recognition of the CMIP5 "hot model" problem. We have not made any changes to the paper, in response to this comment, because: a) we had addressed CMIP5 output extensively in our prior studies; b) the most important aspect of this prior analysis is used in our paper, via the trapezoid.

This is more of a suggestion than a required revision, but I believe the manuscript could benefit from a clearer articulation of the motivation and novel aspect of this research. While the study certainly contributes to the existing literature, particularly on the AR5 and AR6 results (which is sufficient justification in my opinion), it currently lacks a concise statement explaining how this specific analysis advances the field. Adding to the substantial research on AR5 and AR6 outcomes is a valuable addition to the publication record but highlighting why it was necessary to conduct this analysis using EM–GC could elevate the paper and its overall impact.

We have added the following new sentence, to the end of the second paragraph of the Introduction:

The projections of GMST shown throughout this paper are motivated by quantifying the likelihood of achieving either the target (1.5  $^{\circ}$ C) or upper limit (2.0  $^{\circ}$ C) of global warming under the PA, for various assumptions regarding GHGs and aerosols.

**Specific Comments**

L25: Is large ensemble generation a unique capability of EM-GC? The phrasing of this sentence sort of implies that this capability is unique to only EM-GC.

We have not changed this sentence in the abstract, which reads:

The hallmark of our RCM is the ability to conduct large (here, 160,000 member) ensemble forecasts of global warming.

because this sentence highlights an important aspect of the research shown in the paper. The ability to run ensemble members of this size, while not unique to the EM-GC, is certainly not possible to accomplish with any ESM. The text does not state "unique". We have left the text as is because the ability to run truly large ensembles is a "unique" feature of the class of models known as "Reduced Complexity Models", as detailed in Table 1 of Nicholls et al. (2021).

L41: The authors might want to consider a different reference here, while Meinshausen et al., 2020 demonstrates the capability it is not a model documentation manuscript. Perhaps a lit review of RCMs like the RCMIP manuscripts would be more appropriate here. Or consider referencing additional RCMs and using the respective documentation manuscripts instead of only citing a specific model application manuscript.

We have added citations to the RCM papers Nicholls et al., 2020 and Nichols et al., 2021 to line 41.

L54: The text beginning here is a tad unclear and I would like to make sure I am understanding this section correctly - so in the previous study (McBride 2021) EM-GC results were from the SSP scenarios published between AR5 and AR6. But the EM-GC RF parameter values were from AR5. This manuscript uses the AR6 parameterizations for RF and the AR6 scenarios? So essentially this paper is EM-GC uses different equations/parameters for RF and different scenarios compared to McBride?

Table 1 has been added to summarize the model inputs used for the Baseline and AR6 frameworks.

L90: The text "The second number in the name of the SSP scenario is the target RF at the end of the century in units of W m-2, commonly referred to as the "nameplate RF" (O'Neill et al., 2014)" seems a bit out of place and could probably be removed.

We have decided to retain this sentence, as written in the second paragraph of the "Data and methods" section, because the word "nameplate" appears four more times in the paper. For this sentence in question, we feel a definition is important, even though most of the readers might understand the latter context had this sentence been removed.

L109: The phrase "permafrost melt" is informal, I encourage the authors to use "permafrost thaw" instead.

**Change made**

L265 - in the text "Two of these metrics quantify how well the modelled GMST anomaly represents the observed temperature anomaly of the atmosphere for the entire training

period (1850–2019,  $\chi$ 2 ATM) and over the last 80 years (1940 – 2019,  $\chi$ 2RECENT)" is the 1940-2019 temperature observations being counted twice? Is it using the same temperature observations or different ones? Some more details would be helpful here, or a clear justification as to why there are two goodness of fit metrics for temperature anomaly.

L266 - "The  $\chi$ 2RECENT metric is used because without this constraint" what constraint? The  $\chi$ 2 ATM  $\leq$  2 or the  $\chi$ 2 OCEAN constraint?

Here we are replying to the comments on both L265 and L266.

**The sentence that had read:**

The  $\chi^2_{\text{RECENT}}$  metric is used because without this constraint, some solutions with values of  $\chi^2_{\text{ATM}} \leq 2$  have a visually poor simulation of the rise in GMST over the past 4 to 5 decades, due to the large uncertainty associated with early measurements of  $\Delta T$  (McBride et al., 2021).

**has been changed to read:**

The  $\chi^2_{\text{RECENT}}$  metric is used because without this particular constraint, some solutions with values of  $\chi^2_{\text{ATM}} \le 2$  have a visually poor simulation of the observed rise in GMST over the past 4 to 5 decades, due to the large uncertainty associated with early measurements of  $\Delta T$ , as described in Sect. 2.1 of McBride et al. (2021).

**Reviewer 2:**

Farago et al. present an assessment of how the updates to concentrations of GHGs, parametrisations of their effective radiative forcings (ERFs), and tropospheric aerosol ERFs, made between the 5th and 6th IPCC assessment reports, influence projected 21st century warming in a semi-empirical reduced complexity model. They show that, collectively, these changes increase the projected warming substantially. This is an impressive and clearly written study. It makes a useful contribution which I suspect will be of wide interest. I support it's publication. I have suggested some minor changes below, all of which I leave to the discretion of the authors whether to implement.

Thanks for these very kind words. We also sincerely appreciate the constructive comments of this review.

**General comments**

It would be valuable to quantify the relative contributions to warming of the different changes made between the frameworks. It may be that this is not possible to do without significant extra analysis, but I suspect many readers will be left wanting to know the answer, and if only a qualitative sense of the relative contributions could be given, this would still be valuable. More broadly, I think the Conclusions section could benefit from some more qualitative explanation to guide the reader to the right interpretation of what is driving the main findings. There's a lot in this paper, and the reader (or at least this reader) could have benefitted from some help on this point. I assume that one major contributor to the extra warming is the combination of a larger positive GHG forcing and a more negative aerosol forcing during the recent observed period under the AR6 framework (i.e. more aerosol masking of warming under the AR6 framework, with the same total recent warming), which then leads to more rapid warming as aerosols decline and GHGs rise over the coming decades. Do the authors agree?

Thanks for this great suggestion. A new paragraph towards the end of Section 3.2, as well as a new Table 4, has been added to address this important point. The new paragraph reads as follows:

Larger values of  $\Delta T_{2100}$  are found in the AR6 framework compared to the Baseline (Table 2). As detailed in Table 4, the more aggressive warming within the AR6 framework is due mainly to three factors: 1) stronger cooling over the historical time period by tropospheric aerosols in the AR6 framework relative to Baseline (Fig. S1e); 2) larger future concentrations of  $CO_2$  projected by AR6 compared to the SSP database (Fig. S1a); 3) greater ERF due to  $CO_2$  using AR6 formulae compared to the AR5 formulae. Table 4 shows the change in the median value of  $\Delta T_{2100}$ , found for a series of full ensemble model simulations conducted using the AR6 framework, except for replacement of individual model inputs from the Baseline run. The entry labeled " $CO_2$  PPM" for SSP4–3.4 represents the difference

between a computation of  $\Delta T_{2100}$  found using AR6-based model inputs (2.10  $^{\circ}$ C for SSP4-3.4, as shown in Table 2) and a new median value of  $\Delta T_{2100}$  found from a simulation that uses the Baseline projection of the atmospheric concentration of CO2 (dotted line, Fig. S1a) and AR6 values for all other model inputs (in this case, yielding a median value for  $\Delta T_{2100}$  of 1.986  $^{\circ}$ C for SSP4-3.4). Similarly, the entry labeled "CO $_2$  RF Formula" shows the difference in  $\Delta T_{2100}$  from the full AR6 simulation compared to a run that uses the ERF formulae for CO2 from Table 8.SM.1 of AR5. The fact that the single, largest impact on  $\Delta T_{2100}$  is driven by changes to the ERF of tropospheric aerosols, for both the SSP4-3.4 and SSP2-4.5 scenarios, underscores the importance of reducing the current uncertainty in this quantity, to better constrain future projections of global warming. Furthermore, the most important GHG-related factors affecting our forecasts of ΔT2100 are the projections of the future atmospheric concentrations of CO2 and CH4, which are notably different within Annex III of AR6 compared to the SSP database. Finally, changes to other inputs between the AR6 and Baseline computations, that are not considered in Table 4 such as the ERF due to tropospheric ozone, halocarbons, and LUC, make small contributions to the differences in the model projections of  $\Delta T_{2100}$ .

• Given that the changes between AR5 and AR6 frameworks are so fundamental, I wonder if a summary table or figure in the main article in Section 2.1 Model Inputs, to give the reader a quick sense of the most important of these changes, would be useful?

Thanks for this suggestion. A new table, Table 1, has been added to summarize the model inputs used for the Baseline and AR6 frameworks.

**Minor comments:**

**Abstract**

• ".. to the state of knowledge that existed" I am not sure that the pre-AR6 framework is synonymous with the existing knowledge. Perhaps it would be better here to make a limited comparison to e.g. 'the AR5 framework' or similar?

We have left the phrase "we compare and contrast these AR6 updates to the state of knowledge that existed prior to the publication of AR6" in the abstract, because we prefer to not define and then use the term "Baseline" within the abstract. This phrase, perhaps imperfect, reflects our best attempt to succinctly and accurately define the term "Baseline" as used in the paper, without using the word "Baseline" in the abstract.

• "the rate of warming is ... within the AR6 framework" having read the paper, it is clear what this means, but as a standalone sentence it reads strangely. There is only one observed rate of warming over a historical period. Consider rewording – e.g. "the modelled rate of warming". There is a broader point here that the abstract might benefit from a

statement of the relationship between observations, RCMs, and ESMs, to give the non-expert reader a sense of the motivation for the study (i.e. to provide "data-driven probabilistic" projections, as is noted in the Introduction).

The word "modeled" has been added, as suggested.

We have also added "data-driven" in front of "probabilistic" projections on line 17 of the abstract

**1 Introduction**

• Line 55 "our": here and later in the paragraph "we" and "our" etc. are used to refer to the McBride et al., 2021 paper. I found this a little confusing as I expect these terms to imply a statement made from the perspective of this study (even if authors are the same on both studies)

The other reviewer also commented on the over-use of "our", which we have addressed.

The sentence on line 55 that had read:

In the McBride et al. (2021) paper, our projections of GMST

has been changed to read:

In the McBride et al. (2021) paper, their projections of GMST

**2. Model inputs**

Is there any reason not to use "AR5" throughout instead of "Baseline"?

We have decided to retain baseline, rather than "AR5", because the SSP scenarios that define future GHG concentrations were developed after the publication of AR5.

**3.2 Probabilistic forecast on future warming**

• Line 500: this point, that given our updated knowledge, the highest emissions scenario with a good chance of achieving 2°C has essentially moved down one SSP, from 3.4 to 2.6, is I think the single most significant finding of the study, and will be of broad interest. I would suggest adding a sentence stating this to the **abstract**, even if that meant removing some of the other quantitative details given there.

Great point; much thanks. We have added the following last sentence to the abstract:

For society to have high confidence in achieving at least the upper limit of 2  $^{\circ}$ C warming of the PA, the radiative forcing of climate due to GHGs must be placed close to the SSP1–2.6 pathway over the coming decades.